# PSPC1-interchanged interactions with PTK6 and β-catenin synergize oncogenic subcellular translocations and tumor progression

Yaw-Dong Lang [1], Hsin-Yi Chen[2,3], Chun-Ming Ho[1,4,5], Jou-Ho Shih[1], En-Chi Hsu [1], Roger Shen[1,6], Yu-Ching Lee[7], Jyun-Wei Chen[1], Cheng-Yen Wu[1], Hsi-Wen Yeh [1], Ruey-Hwa Chen [8] & Yuh-Shan Jou [1,5,6]*

Hepatocellular carcinoma (HCC) is one of the most lethal cancers worldwide due to metastasis. Paraspeckle component 1 (PSPC1) upregulation has been identified as an HCC pro-metastatic activator associated with poor patient prognosis, but with a lack of targeting strategy. Here, we report that PSPC1, a nuclear substrate of PTK6, sequesters PTK6 in the nucleus and loses its metastasis driving capability. Conversely, PSPC1 upregulation or PSPC1-Y523F mutation promotes epithelial-mesenchymal transition, stemness, and metastasis via cytoplasmic translocation of active PTK6 and nuclear translocation of β-catenin, which interacts with PSPC1 to augment Wnt3a autocrine signaling. The aberrant nucleocytoplasmic shuttling of active PTK6/β-catenin is reversed by expressing the PSPC1 C-terminal interacting domain (PSPC1-CT131), thereby suppressing PSPC1/PTK6/β-catenin-activated metastasis to prolong the survival of HCC orthotopic mice. Thus, PSPC1 is the contextual determinant of the oncogenic switch of PTK6/β-catenin subcellular localizations, and PSPC1-CT131 functions as a dual inhibitor of PSPC1 and PTK6 with potential for improving cancer therapy.

[1] Institute of Biomedical Sciences, Academia Sinica, Taipei 11529, Taiwan. [2] Graduate Institute of Cancer Biology & Drug Discovery, College of Medical Science & Technology, Taipei Medical University, Taipei 11031, Taiwan. [3] Ph.D. Program for Cancer Molecular Biology and Drug Discovery, College of Medical Science and Technology, Taipei Medical University, Taipei 11031, Taiwan. [4] Institute of Bioinformatics and Systems Biology, National Chiao Tung University, Hsin-Chu 30068, Taiwan. [5] Bioinformatics Program, Taiwan International Graduate Program, Institute of Information Science, Academia Sinica, Taipei 11529, Taiwan. [6] Program in Molecular Medicine, National Yang-Ming University and Academia Sinica, Taipei 11221, Taiwan. [7] TMU Research Center of Cancer Translational Medicine, Taipei Medical University, Taipei 11031, Taiwan. [8] Institute of Biological Chemistry, Academia Sinica, Taipei 11529, Taiwan. *email: jou@ibms.sinica.edu.tw

Cancer cell dissemination and colonization at distant organs that overwhelms essential homeostasis of the organism is the major cause of death of cancer patients[1,2]. Metastasis is a complicated process with uncontrollable outgrowth of reprogrammed cancer cells spreading into the bloodstream for colonization and establishment of a pro-metastatic niche at distant organs followed by reformation of the extracellular matrix, alteration of vascular supply, increased pro-metastatic cytokines, and interaction with stromal cells in the tumor microenvironment[3]. Therefore, targeting prognosis-related metastasis driver genes with a detailed understanding of the pathological mechanisms is critical to eliminating the progressive signaling in the metastatic process and prolonging the life of patients.

We recently discovered that upregulation of PSPC1 is associated with poor patient survival in cancers, including human hepatocellular carcinoma (HCC)[4], a predominant form of liver cancer occurring mainly in developing countries. PSPC1 enhances epithelial–mesenchymal transition (EMT), stemness and tumor growth through the activation of core transcription factors (TFs), such as EMT-TFs (e.g., Snail, Slug, and Twist), cancer stem-like cells (CSC)-TFs (e.g., Oct4, Nanog, and Sox2), and c-Myc[4]. PSPC1 is also the interacting partner of Smad2/3 acting as a contextual determinant of TGF-β1 responses to switch the dichotomous TGF-β1 function from tumor suppressing in precancerous cells to pro-metastatic signaling in malignant cancer cells[4,5].

To explore detailed PSPC1 regulatory mechanisms, we performed protein immunoprecipitation (IP)/liquid chromatography mass spectrometry (LC–MS)/MS analysis and identified a non-receptor protein tyrosine kinase 6 (PTK6) as the PSPC1-interacting partner. Similar to TGF-β1, PTK6 also exerts dual roles in tumorigenesis. Overexpression of PTK6 is found in late tumor stages and is associated with poor patient prognosis in multiple cancer types including HCC[6–8]. However, its high expression is also found in nondividing epithelial cells suppressing cell proliferation and promoting differentiation and apoptosis[9]. Emerging evidence has revealed that such dual functions of PTK6 might be associated with its subcellular localizations. In the nucleus, PTK6 phosphorylates the RNA-binding protein Sam68[10], which suppresses PTK6-induced cell proliferation[11]. PTK6 also phosphorylates the splicing factor PSF to induce PSF cytoplasmic relocalization and cell cycle arrest[12]. Conversely, cytoplasmic PTK6 phosphorylates more than 30 intracellular targets to promote oncogenic functions[7,13,14]. Consistent with this notion, nuclear PTK6 is found in well-differentiated prostate cancers but is absent in poorly differentiated tumors[15]. Furthermore, forced localization of PTK6 to the plasma membrane or nucleus results in cell proliferation or growth inhibition, respectively[16]. Therefore, the function of PTK6 seems to depend on the context of cell type, differentiation state, and intracellular localization.

Aberrant nucleocytoplasmic shuttling of oncogenic and tumor suppressor proteins is known to contribute to tumor progression[17,18]. Prominent examples, include p53[19], survivin[20], FOXO[21], β-catenin[22], and PTK6[12], whose localization to the wrong subcellular compartment results in tumor progression in various human cancer types[23]. Thus, pharmacological agents that can correct the aberrant subcellular compartmentalization have recently emerged as a class of anticancer drugs. For instance, selinexor (also called KPT-330), a small molecule inhibitor for the nuclear exporting factor CRM1 (chromosome region maintenance 1 protein, also known as exportin 1 or XPO1), has been demonstrated to induce nuclear accumulation of tumor suppressor proteins and antitumor activities in preclinical models and human clinical trials[24].

Here, we reveal that PSPC1 upregulation is the contextual determinant of the oncogenic switch of PTK6 translocation from the nucleus to the cytoplasm and of β-catenin nuclear translocation. PSPC1 upregulation or PSPC1-Y523F mutation loses nuclear sequestration of tumor suppressive PTK6 to promote cancer EMT, stemness and metastasis via oncogenic cytoplasmic translocation of PTK6 and nuclear translocation of β-catenin, which interacts with PSPC1 to promote Wnt3a autocrine signaling. We further identify PSPC1-CT131 as an inhibitor to abrogate the oncogenic functions of PSPC1 and PTK6 and to interfere with oncogenic translocations of PTK6 and β-catenin, thus suppressing tumor progression in HCC models.

## Results

**PSPC1 interacts with PTK6 to inhibit tumor progression**. To identify PSPC1-interacting proteins, we performed PSPC1 IP with Huh-7 cell lysate, separated the co-precipitated proteins by gel electrophoresis (Supplementary Fig. 1a, top), and subjected them to LC–MS/MS analysis. In addition to known partners, such as paraspeckle proteins PSF, PSPC1 and p54nrb, we identified PTK6[8] as a putative PSPC1-interacting protein (Supplementary Fig. 1a and Supplementary Table 1). IP/Western blotting analysis further confirmed the interaction of endogenous or ectopic PSPC1 with PTK6 in HCC cells lines, such as Huh-7 and SK-hep1 (Fig. 1a–c).

Since PTK6 is a nonreceptor tyrosine kinase, we next determined whether the PSPC1 and PTK6 interaction is direct and through a tyrosine phosphorylation-dependent manner. We performed an in vitro pull-down assay with recombinant GST-PSPC1 (Supplementary Fig. 1b) and IP/Western blotting analysis with the phosphotyrosine-specific antibody 4G10 (Supplementary Fig. 1c). We found that PSPC1 and PTK6 interacted directly in vitro. Furthermore, the interaction of PSPC1 with PTK6 was disrupted by expression of the PTK6 kinase-dead mutant, PTK6-KM (PTK6-K219M), or treatment of the IP complex with a nonspecific calf-intestinal alkaline phosphatase (CIP), suggesting a role of PTK6-induced tyrosine phosphorylation in their interaction (Supplementary Fig. 1c, d).

Since PTK6 subcellular localization might determine its tumorigenic functions, we monitored the subcellular localizations of PTK6 and PSPC1 by immunofluorescence (Supplementary Fig. 1e) and subcellular fractionation (Fig. 1d). While the inactive PTK6-KM was mostly present in the cytoplasm, the active PTK6 (p-PTK6, detected by anti-pY342 PTK6 antibody) was enriched in the nucleus where it was partially colocalized with nuclear-localized PSPC1 (Fig. 1d and Supplementary Fig. 1e). These data are consistent with the notion that PSPC1 interacts with active PTK6 in the nucleus. Domain mapping analysis indicated that the C-terminal proline-rich domain of PSPC1 and the SH2 and SH3 domains of PTK6 were essential for their interaction (Supplementary Fig. 1f, g). Given the interaction of SH2 domain with phosphotyrosine, our data are consistent with the binding of PTK6 with tyrosine-phosphorylated PSPC1.

PSPC1 is known to act as an pro-metastatic driver in HCC[4], whereas nuclear PTK6 is implicated in tumor suppression[7]. To determine the tumorigenic impact of the PSPC1 and PTK6 interaction in HCC cells, we overexpressed PSPC1 and PTK6 (or PTK6-KM) in SK-hep1 cells (Fig. 1e), which expressed no PSPC1 and a low level of PTK6 (Supplementary Fig. 1h). We found that PTK6, but not PTK6-KM, diminished PSPC1-enhanced cell migration and invasion (Fig. 1f, g). Conversely, knockdown of PTK6 with different short hairpin RNAs (shPTK6-#52 and -#53) in SNU-387 cells, which expressed no PSPC1 and a high level of PTK6 (Supplementary Fig. 1h), abolished cell migration and invasion in PSPC1 overexpression conditions (Fig. 1h–j). Together, our data suggest that nuclear PTK6 forms a complex with tyrosine-phosphorylated PSPC1, which suppresses the tumor progression effects of PSPC1.

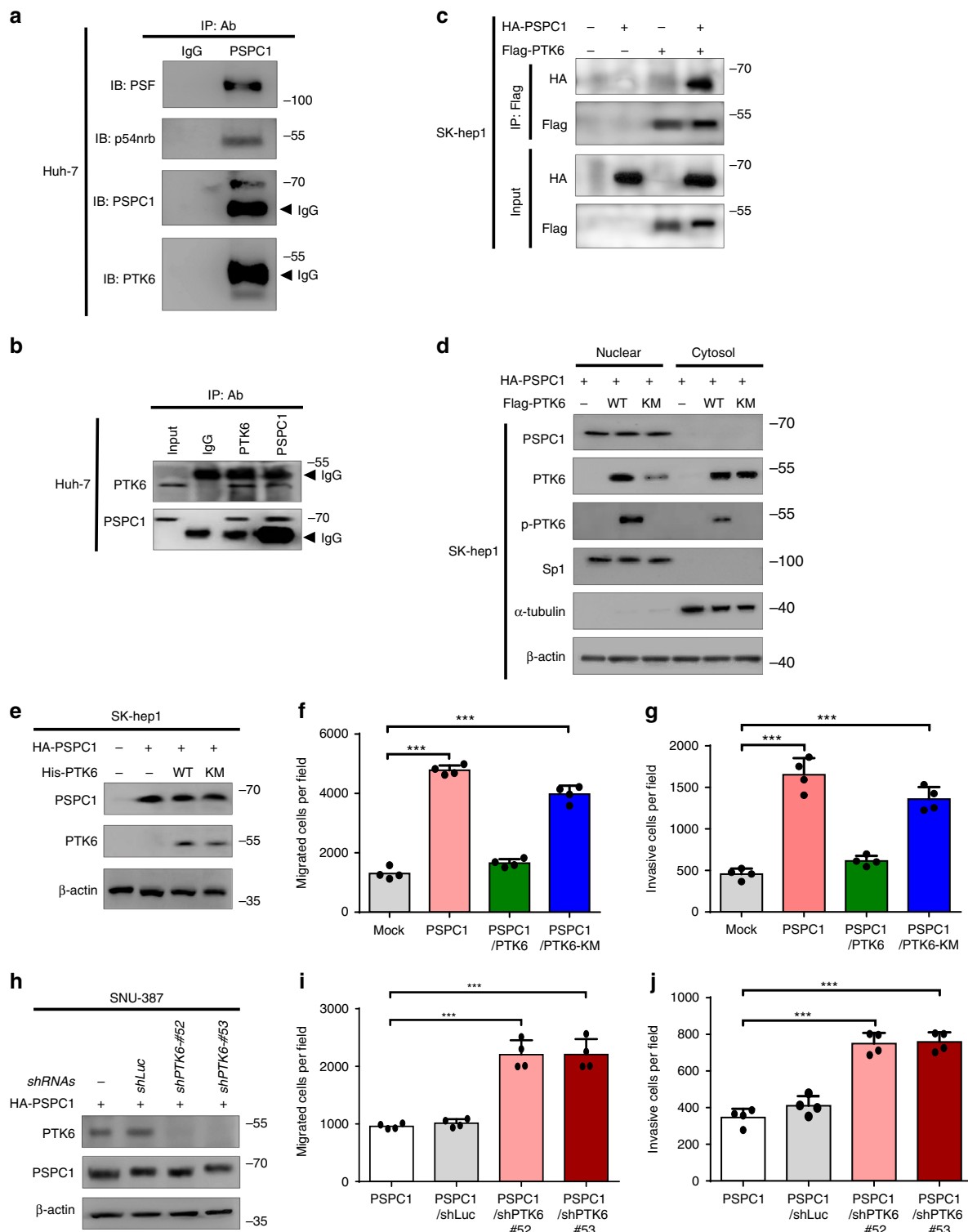

**PSPC1-Y523F releases PTK6 to a synergism in tumorigenicity**. We sought to identify the phosphotyrosine residue on PSPC1 critical for PTK6 interaction to validate the decisive role of PSPC1/PTK6 interaction in HCC tumorigenicity. Two tyrosine residues Y383 and Y523 located within or near the PTK6-interacting region of PSPC1, i.e., the C-terminal domain of PSPC1, were substituted with phenylalanine (PSPC1-Y383F and PSPC1-Y523F) for structural simulation and functional evaluation (Fig. 2a, top). Our results showed that the PSPC1-Y383F and PSPC1-Y523F mutations had similar protein structural folding compared to wild-type PSPC1 (Supplementary Fig. 2a, b). Nevertheless, IP/Western blotting analysis indicated that Y523F, but not Y383F, abrogated the interaction with PTK6 (Fig. 2a). Three-dimension structural docking analysis revealed that most of the hydrogen bonds near the PSPC1 Y523 residue were involved in contacting PTK6 (Supplementary Fig. 2c). Moreover, an in vitro kinase assay followed by Western blotting with 4G10 anti-phosphotyrosine antibody demonstrated that recombinant human PTK6 phosphorylated PSPC1, but not PSPC1-Y523F mutant (Supplementary Fig. 2d). These findings indicate that

**Fig. 1 PSPC1 interacts with nuclear tumor suppressive PTK6 to inhibit tumor progression. a** The PSPC1 immunoprecipitated (IP) complex of Huh-7 cells lysates with anti-PSPC1 antibody analyzed by Western blotting (IB) with the indicated antibodies (IP/Western). **b** Endogenous PSPC1 and PTK6 interaction analyzed by IP/Western blotting analysis in Huh-7 cells. Preimmune IgG is the control, and the arrowhead is the IgG heavy chain. **c** Interaction of PSPC1 with PTK6 by ectopic expression of HA-tagged PSPC1 and/or Flag-tagged PTK6 protein in SK-hep1 cells by IP/Western analysis. **d** Subcellular distribution of wild-type (WT) and kinase-dead (KM) mutant of flag-tagged PTK6 in SK-hep1 cells expressing HA-tagged PSPC1 determined by Western blotting analysis. Sp1 and α-tubulin were used as internal controls for nuclear and cytoplasmic fractions, respectively. **e–g** PTK6 suppresses PSPC1-enhanced cell motility in a phosphorylation-dependent manner. The expression levels of HA-tagged PSPC1, His-tagged PTK6 wild-type (WT) or kinase dead (KM) mutant in SK-hep1 cells analyzed by Western blotting analysis (**e**). Expression of PTK6, but not KM mutant of PTK6, suppressed PSPC1-mediated cell migration (**f**) and invasion (**g**) in SK-hep1. Data are represented as mean ± SEM ($n = 4$). **h–j** PTK6 suppresses PSPC1-enhanced cell motility reversed by PTK6 knockdown. PTK6 knockdown efficiency of shRNAs (*shRNA#52 and shRNA#53*) and Western blotting analysis in SNU-387 cells expressing HA-tagged PSPC1 (**h**). PTK6 knockdown increased cell migration (**i**) and invasion (**j**) in SNU-387 cells. *shLuc* is a control shRNA targeting luciferase. Data are represented as mean ± SEM ($n = 4$). All data statistics based on: *$p < 0.05$, **$p < 0.01$, ***$p < 0.001$ by one-way ANOVA with Brown–Forsythe test. Source data are provided as a Source Data file.

PTK6 phosphorylates the Y523 residue of PSPC1, which promotes the binding of PSPC1 to PTK6.

To dissect the functional consequences of PSPC1 phosphorylation by PTK6, we expressed the phosphorylation-defective mutant PSPC1-Y523F to test its effect on the tumor suppressive functions of the PTK6/PSPC1 axis, and PSPC1-Y383F was included as a control. As expected, PTK6 suppressed PSPC1-mediated cell migration and invasion in PSPC1/PTK6 and PSPC1-Y383F/PTK6 SK-hep1 transfectants (Fig. 2b, c). In contrast, PSPC1-Y523F was refractory to the inhibitory effects of PTK6 and further enhanced cell migration and invasion in PSPC1-Y523F and PSPC1-Y523F/PTK6 SK-hep1 transfectants (Fig. 2b, c). PSPC1-Y523F showed a similar enhancement of cell motility in PTK6 highly expressed SNU-387 cells, compared with wild-type PSPC1 or PSPC1-Y383F (Fig. 2d, e). These results support a role for PTK6-induced PSPC1 phosphorylation at Y523 in the suppression of HCC cell motility, and disruption of this phosphorylation reverses this tumor suppression effect.

The ability of PSPC1-Y523F to revert the tumor suppressive function of the PTK6/PSPC1 axis raised a hypothesis that this mutant might promote the shuttling of PTK6 from nucleus to cytoplasm. Indeed, we found that the total and active PTK6 were expressed in the nucleus when coexpressed with PSPC1 (Fig. 2f). However, when coexpressed with PSPC1-Y523F, the total and active PTK6 could be found in cytoplasm and on plasma membrane (Fig. 2f). Meanwhile, we expressed PTK6 in PSPC1-deficient SK-hep1 cells and detected subcellular localizations of total PTK6 and p-PTK6 to clarify the role of PSPC1 in PTK6 subcellular translocation. We found that total PTK6 might distribute evenly in different subcellular fractions but p-PTK6 is expressed more focusing on the locations of cytoplasm and cell membrane (Supplementary Fig. 3a). Therefore, the alteration of PTK6 subcellular localization implied its functional switch is depending on PSPC1 sequestration.

Of note, previous studies showed that plasma membrane-targeted PTK6 promotes β-catenin-regulated transcription[25] and EMT[26], whereas nuclear-targeted PTK6 inhibits β-catenin-regulated transcription[25] and cell growth[16]. Since PSPC1 also elicits EMT and stemness functions[4], we postulate that the PSPC1-Y523F-induced redistribution of PTK6 could not only switch the function of PTK6 from tumor suppression to tumor promotion but also facilitate a synergism between the oncogenic PTK6 and PSPC1-Y523F. We thus tested this synergism in tumor promotion by coexpression of PTK6 and PSPC1-Y523F in SK-hep1 cells or by expressing PSPC1 or PSPC-1-Y523F in PTK6 highly expressed SNU-387 cells. The synergism of PSPC1-Y523F with exogenous or endogenous PTK6 in various tumor-promoting functions was demonstrated in SK-hep1 or SNU-387 cells, respectively, including the increase in EMT morphology, upregulation of the mesenchymal marker

N-cadherin, downregulation of the epithelial marker γ-catenin, increased expression and promoter activity of EMT-TFs (Snail and Slug), elevated expression of CSC-TFs (Nanog, Sox-2 and Oct-4) and increased stemness features as evidenced by spheroid formation and side population assays (Fig. 2g–j and Supplementary Fig. 3b–e). Together, our results suggest that the PSPC1-Y523F mutation releases PTK6 from nuclear sequestration to facilitate its cytoplasmic translocation, which facilitates a functional synergism between cytoplasmic PTK6 and PSPC1-Y523F to enhance tumorigenic features, such as EMT, motility, and stemness.

**PSPC1 modulates Wnt signaling and Wnt3a autocrine function**. To elucidate the underlying mechanism of the roles of PSPC1/PTK6 reciprocal modulation in tumorigenesis, we selected hepatocyte growth factor (HGF)/c-Met-treated Huh7 cells as a model to modulate the PSPC1/PTK6 axis[27,28] and to examine its effect on the Wnt/β-catenin pathway downstream of PTK6[25,29]. As expected, HGF increased cell mobility and induced EMT by switching expression from E-cadherin to N-cadherin (Fig. 3a and Supplementary Fig. 4a, b). More importantly, HGF stimulation increased the nuclear levels of PSPC1 and β-catenin but reduced those of total and active PTK6 (p-PTK6), which was accompanied by decreased β-catenin and increased p-PTK6 in cytoplasm and on plasma membrane (Fig. 3a). Consistent with the increased levels of nuclear β-catenin and cytoplasmic/membrane p-PTK6, HGF stimulation increased the PSPC1/β-catenin interaction but reduced the PSPC1/PTK6 interaction (Fig. 3b).

To investigate the detailed mechanisms of HGF-activated oncogenic p-PTK6, we performed RNA sequencing (RNA-Seq, GSE114856) analysis of untreated and HGF-treated Huh-7 cells and validated the expression of several altered genes by quantitative RT-qPCR (Supplementary Fig. 4c and Supplementary Table 2). Gene sets enrichment analysis (GSEA) of altered genes revealed the enrichment of not only oncogenic profiles such as EMT, metastasis and stemness but also activated c-Met signaling pathway (Supplementary Fig. 4d). IP/Western blotting analysis demonstrated the interaction of c-Met with PTK6 with or without the treatment of c-Met inhibitor XL-184. XL-184, however, blocked the HGF-induced PTK6 Y342 phosphorylation, suggesting a role of c-Met kinase activity in promoting the phosphorylation of PTK6 at Y342 to increase its oncogenic functions (Supplementary Fig. 4e, f).

Given that HGF-upregulated PSPC1 showed a decreased interaction with PTK6 but an increased interaction with β-catenin, we hypothesized that PSPC1-Y523F might similarly favor an interaction with β-catenin but not PTK6, thereby potentiating Wnt/β-catenin signaling. Consistent with this idea, expression of PSPC1-Y523F in SK-hep1 cells carrying exogenous PTK6 or SNU-387 cells carrying endogenous PTK6 increased β-catenin and reduced p-PTK6 in the nucleus with a concomitant

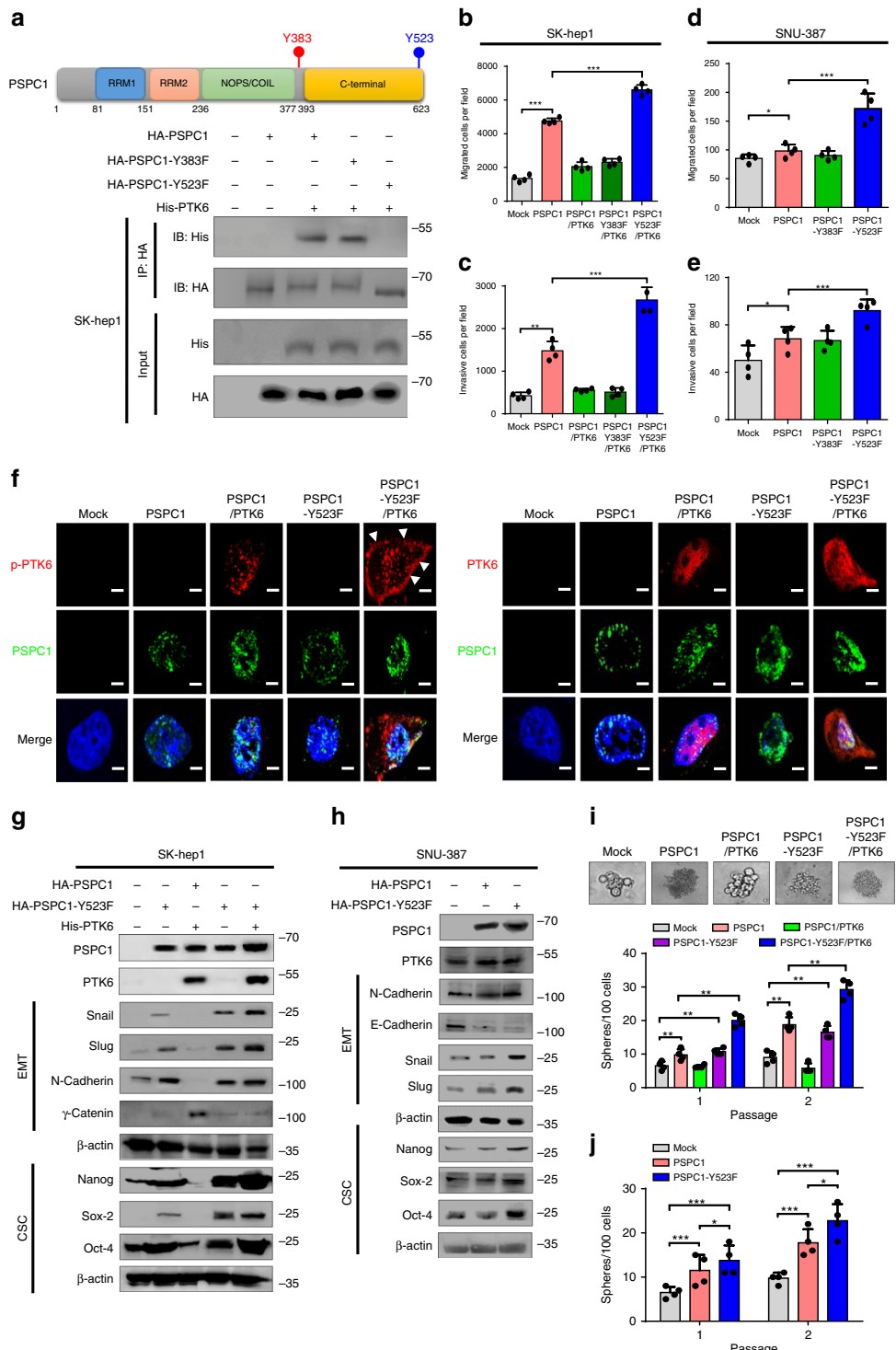

increase in cytosolic and membrane-bound p-PTK6, whereas expression of wild-type PSPC1 had weaker or no effects (Fig. 3c–e and Supplementary Fig. 4g). More importantly, compared with that of wild-type PSPC1, PSPC1-Y523F expression showed increased binding to nuclear β-catenin and decreased binding to PTK6 (Fig. 3d). Our results indicated that PSPC1-Y523F expression shifts the preference of its nuclear partner from PTK6 to β-catenin, which likely contributes to the functional synergism between PSPC1 and cytoplasmic PTK6 in tumor promotion.

Interestingly, using the TOP-Flash/FOP-Flash luciferase assay to evaluate the transcription-activating function of β-catenin, we found that PSPC1 overexpression in 293T cells readily induced reporter activity, similar to the effect of constitutively active β-catenin (β-catenin del-N mutant)[30] and Wnt3a but not Wnt1. Furthermore, knockdown of PSPC1 impaired the transactivation function of the β-catenin del-N mutant, whereas expression of a dominant-negative TCF4 mutant (TCF-DN), which disrupts TCF4 and β-catenin interaction[31], abrogated the effect of PSPC1 on the activation of reporter activity (Fig. 3f). These data identified a role of PSPC1 in promoting β-catenin-mediated canonical Wnt signaling, most likely through its interaction with β-catenin. Consistent with this notion, expression of PTK6, which

**Fig. 2 PSPC1 or PSPC1-Y523F releases PTK6 to synergize the oncogenic effects. a** The PSPC1-Y523F but not PSPC1-Y383F mutation abolished the PSPC1/PTK6 interaction demonstrated by IP/Western blotting analysis. (Top) A cartoon illustrates domain structures and tyrosine residues Y383 (Red) and Y523 (Blue) in the C-terminal PSPC1 protein. **b, c** PTK6 suppressed the PSPC1-enhanced cell migration (**b**), invasion (**c**), in SK-hep1 cells. Mutation of PSPC1-Y523F but not Y383F abolished the PSPC1-enhanced cell migration and invasion. Data are represented as mean ± SEM ($n = 4$). **d, e** SNU-387 expressed a higher level of PTK6 could to some extent reduce PSPC1 and the PSPC1-Y383F-enhanced cell migration (**d**), invasion (**e**), but not that of PSPC1-Y523F (loss of PTK6 interaction ability). Data are represented as mean ± SEM ($n = 4$). **f** Immunofluorescence for the detection of subcellular localizations of PSPC1, p-PTK6, and PTK6 in SK-Hep1. Colors for p-PTK6 as well as PTK6, PSPC1, and DAPI are red, green, and blue, respectively. The scale bar represents 10 µm. **g** Western blotting analysis of EMT and cancer stemness markers in SK-hep1 cells expressing PSPC1, PSPC1-Y523F, and PTK6. **h** Western blotting analysis of EMT and cancer stemness markers in SNU-387 cells expressing PSPC1, PSPC1-Y523F. **i** Spheroids formation in different PSPC1/PTK6 SK-hep1 transfectants including phase images (Top) and graphic quantifications after two passages of spheroids (bottom). Data are represented as mean ± SEM ($n = 6$). **j** SNU387 cells expressing higher level of PTK6 could reduce PSPC1-mediated spheroids formation, but cells expressing PSPC1-Y523F could not. Data are represented as mean ± SEM ($n = 6$). PTK6 suppressed PSPC1-enhanced cell spheroids formation in SK-hep1 cells. The PSPC1-Y523F mutation but not Y383F potentiated spheroids formation with series dilutions for 20 days cultures. Co-expression of PSPC1-Y523F and PTK6 in SK-hep1 cells synergized oncogenic spheroid formation. All data statistics based on *$p < 0.05$, **$p < 0.01$ ***$p < 0.001$ calculated by one-way ANOVA with Brown–Forsythe test. Source data are provided as a Source Data file.

blocked the PSPC1 interaction with β-catenin, inhibited the function of PSPC1 in activating β-catenin pathway, whereas co-expression of PSPC1-Y523F/PTK6 in SK-hep1 cells, which led to increased β-catenin binding, showed an even higher reporter activity (Fig. 3g). Finally, we evaluated the possible effect of the PSPC1/PTK6 axis on promoting autocrine Wnt signaling by measuring Wnt3a protein expression in the conditioned medium (CM) of SK-hep1 transfectants. PSPC1 was capable of inducing Wnt3a secretion, and this effect was blocked by PTK6 co-expression. PSPC1-Y523F, however, was insensitive to the inhibitory effect of PTK6 and showed a substantial increase in Wnt3a secretion to establish autocrine Wnt3a/β-catenin signaling (Fig. 3h). Collectively, our results suggested that the dynamic interaction of PSPC1 or PSPC1-Y523F with PTK6 or β-catenin modulated autocrine Wnt3a/β-catenin signaling to synergize tumor progression in HCC.

**The PSPC1/PTK6/β-catenin axis is necessary for metastasis.** To investigate the tumorigenic effects of the PSPC1/PTK6/β-catenin dynamic interaction in vivo, we constructed a set of SK-hep1 transfectants expressing luciferase (Luc) and injected them into mouse livers to establish an HCC orthotopic mouse model[32]. Our results demonstrated that PTK6 expression suppressed PSPC1-induced liver tumor growth (Fig. 4a, b) and metastasis to lung tissues (Fig. 4c, d) as measured by bioluminescence imaging (BLI) intensity. In contrast, PTK6 could not suppress PSPC1-Y523F-induced tumor formation and metastasis and even enhanced these effects. The formation of primary tumor nodules on the surface of liver and metastatic nodules on that of lung was also confirmed by gross analysis and histology (Fig. 4e–h). Moreover, immunohistochemistry (IHC) staining of primary and metastatic tumors showed that PTK6 reversed the effects of PSPC1 on downregulating E-cadherin and upregulating vimentin, β-catenin, c-Myc and Wnt3a. Conversely, PTK6 could not suppress and even synergized the oncogenic effects of PSPC1-Y523F (Supplementary Fig. 5a, b). Together, our results demonstrated that the synergized PSPC1/PTK6/β-catenin axis regulates tumor migratory ability and malignant phenotype leading to tumor growth and metastasis in a mouse orthotopic HCC model.

**Phospho-Y523 PSPC1 is associated with better prognosis.** To relevant natural existence of synergized PSPC1 and PTK6 oncogenic effects of the dynamic PSPC1/PTK6/β-catenin/Wnt3a axis in HCC, we generated an antibody against PSPC1-Y523 phosphorylation by using a phospho-peptide and validated its specificity (Supplementary Fig. 6a–c). We then performed IHC assays on 215 human HCC tissue samples to evaluate the expression of PSPC1, p-Y523-PSPC1, PTK6 and Wnt3a (Fig. 5a and Supplementary Table 3). Our data showed that PSPC1 and Wnt3a had higher expression (H-score with a discriminatory threshold higher than 200) at late (III and IV) stages compared to early (I and II) stages of HCC tumor tissues, whereas the expression levels of nuclear p-Y523-PSPC1 and nuclear PTK6 were higher at early stages than late stages of HCC tumors (Fig. 5b–e). Furthermore, high expression of p-Y523-PSPC1 in patients was correlated with low expression of PSPC1. Accordingly, increased expression of nuclear PTK6 in patients was negatively associated with the expression of PSPC1 in tumors (Fig. 5f). Furthermore, high expression of phospho-Y523-PSPC1 and nuclear PTK6 in the IHC analysis of human HCC samples was associated with better patient survival compared to low expression of either one of them, as demonstrated by Kaplan–Meier analysis (Fig. 5g, h, $p = 0.0021$ and $p = 0.028$, respectively). Taken together, our IHC results on HCC tissues demonstrated that decreased expression of p-Y523 PSPC1 as a marker for tumor progression might be used as an index of therapeutic intervention to improve the survival of HCC patients.

**The PSPC1-CT131 is a dual inhibitor of PSPC1 and PTK6.** Since the C-terminal proline-rich region of PSPC1 could serve as a molecular docking target for PSPC1 and PTK6 (Fig. 6a and Supplementary Fig. 2c and 7a), we hypothesized that this 131-residue region (PSPC1-CT131) might simultaneously target to PSPC1 and SH3 domain[33] of PTK6 to act as a dual inhibitor of PSPC1 and PTK6 to suppress their synergized oncogenic signaling pathway. To determine the tumor suppressive effects of PSPC1-CT131, we constructed and ectopically expressed the PSPC1-CT131-EGFP fusion protein and its nuclear targeting mutant Mut-NLS-CT131 by disrupting the nuclear localization sequence (NLS) in HCC cell line Mahlavu (Supplementary Fig. 7b). We found that PSPC1-CT131 colocalized with and sequestered p-PTK6 in the nucleus, whereas Mut-NLS-CT131 relocated p-PTK6 to the cytoplasm (Fig. 6b, c and Supplementary Fig. 7c). Expression of PSPC1-CT131 but not Mut-NLS-CT131 reduced migration, invasion, spheroids formation (Fig. 6d–f), and EMT features such as diminished N-cadherin and increased E-cadherin expression (Supplementary Fig. 7d). Furthermore, expression of PSPC1-CT131 but not Mut-NLS-CT131 decreased the expressions of PSPC1, cytosolic p-PTK6 and nuclear β-catenin, which was accompanied by increased sequestration of p-PTK6 in the nucleus (Fig. 6g and Supplementary Fig. 7e, f). Our results also showed that PSPC1-CT131 interacted with PSPC1, PSPC1-Y523F, and p-PTK6, but not β-catenin (Supplementary Fig. 7e–g). In addition, PSPC1-CT131 but not Mut-NLS-CT131 reduced Wnt3a and TGF-β1 autocrine signaling, as evidenced by

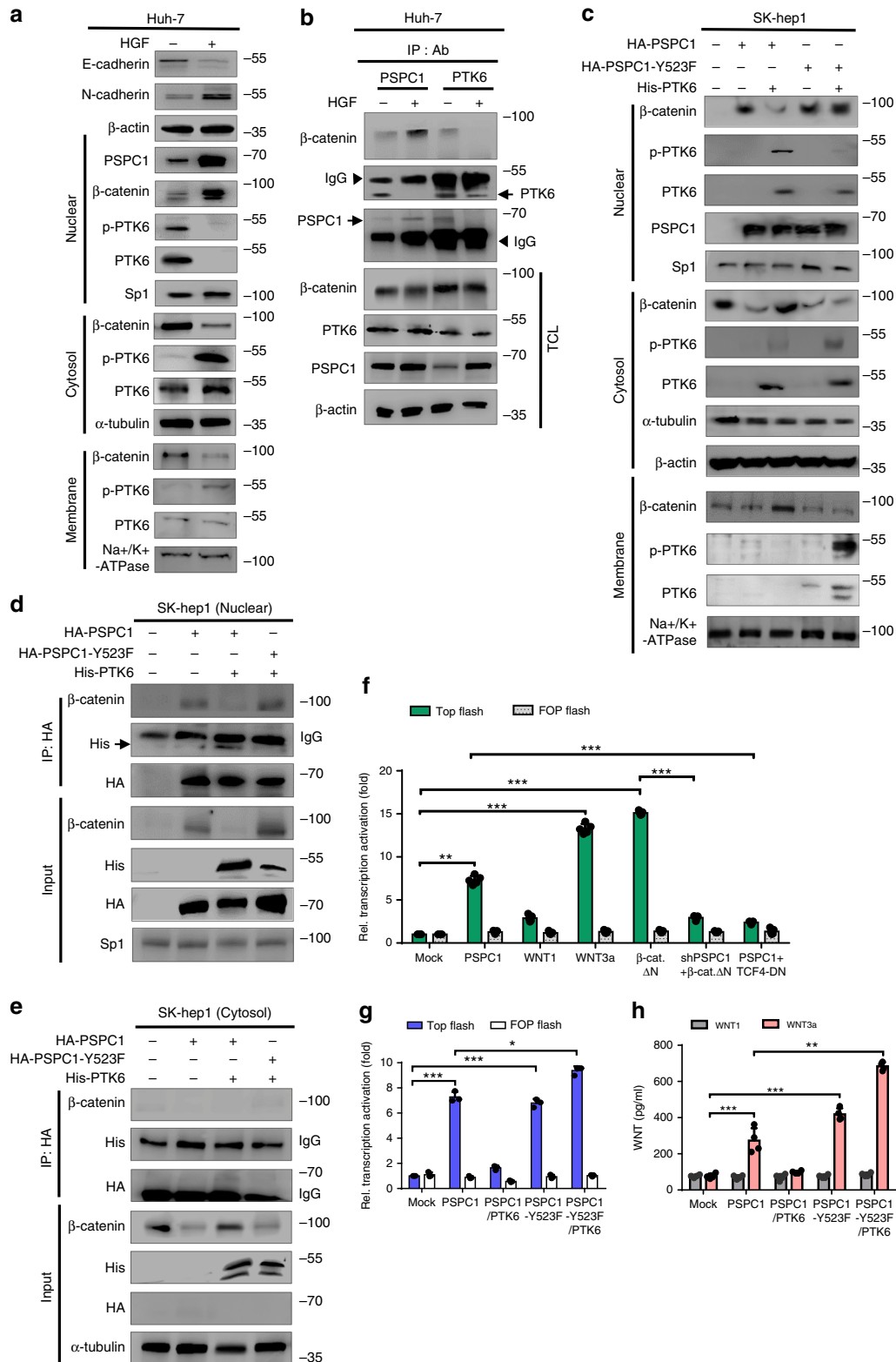

their concentration in the CM (Supplementary Fig. 7h). Collectively, these results demonstrated that PSPC1-CT131 could interact with PSPC1, PSPC1-Y523F, and p-PTK6 in the nucleus to abrogate their synergized functions in tumor progression.

To further elucidate the effect by which PSPC1-CT131 antagonizes the functional synergism of the PSPC1/PTK6 axis, we performed RNA-Seq (GSE114856) analysis on a set of

SK-hep1 transfectants and then analyzed the differentially expressed genes in the gene signatures of GSEA. We selected 30 tumor progression-enriched gene signatures downstream of PSPC1 and PTK6 signaling pathways[4,7,34–38] (Supplementary Table 2) to demonstrate the significant up-regulation (in red) and down-regulation (in green) of gene expression under PSPC1/PTK6 dynamic interaction and PSPC1-CT131 treatment. We

**Fig. 3 PSPC1 modulates Wnt signaling and Wnt3a autocrine function. a** HGF (10 ng/ml) treatment of Huh-7 cells as an EMT induction model upregulates PSPC1 and facilitates reciprocal subcellular translocations of p-PTK6 and β-catenin. The expression of Sp1, α-tubulin and Na+/K+-ATPase, markers of nuclear, cytosolic and membrane localization, respectively. **b** Under HGF stimulation, nuclear PSPC1 interacted with β-catenin and not PTK6 owing to increased cytosolic and membrane active PTK6 expression shown by IP/Western blotting analysis. Arrows indicate proteins after IB. Arrowheads are the IgG heavy chain. **c** Expression of PSPC1-Y523F mutant in SK-hep1 cells decreased nuclear p-PTK6 sequestration to facilitate cytoplasmic and membrane p-PTK6 translocation and facilitate nuclear translocation of β-catenin demonstrated by Western blotting analysis. **d, e** Expression of PSPC1-Y523F mutant reduced the p-PTK6 nuclear sequestration and enhanced PSPC1-Y523F interaction with nuclear β-catenin demonstrated by IP/Western blotting analysis in SK-hep1 cells. **f** Relative transcriptional activity of TOP-Flash and FOP-Flash with TCF4/LEF1 luciferase reporter assays in 293T cells cotransfected with mock control, PSPC1, Wnt3a, β-catenin del-N (deletion of N-terminus), PSPC1 shRNA (shPSPC1)/β-catenin del-N, and PSPC1/TCF4-DN (dominant negative). Data represent the mean ± SEM ($n = 3$). **g** Relative transcriptional activation of TCF4/LEF1 luciferase promoter reporter assays with TOP-Flash or FOP-Flash expressing SK-hep1 cells transfected with vector alone (Mock), PSPC1, PSPC1/PTK6, PSPC1-Y523F and PSPC1-Y523F/PTK6 constructs. Data represent the mean ± SEM ($n = 3$). **h** Secreted cytokines Wnt1 and Wnt3a measured by ELISA assays in the concentrated conditioned medium of SK-hep1 cells transfected with different PSPC1/PTK6 expression constructs. Data represent the mean ± SEM ($n = 3$). All data statistics were based on *$p < 0.05$, **$p < 0.01$, and ***$p < 0.001$ calculated by one-way ANOVA with Brown–Forsythe test. Source data are provided as a Source Data file.

found that the tumor progression gene signatures associated with metastasis, stemness, and the C-Myc, TGF-β1, Wnt/β-catenin, and PTK6 oncogenic pathways were significantly upregulated in cells expressing the PSPC1, PSPC1-Y523F, and PSPC1-Y523F/PTK6 constructs (Fig. 6h). In contrast, these tumor progression signatures were significantly downregulated in PSPC1/PTK6- and PSPC1-CT131-treated HCC cells (Fig. 6h). We further validated the dose-dependent downregulation of oncogenic core proteins such as EMT-TFs, CSC-TFs, C-Myc, β-catenin, p-SMAD2/3, and p-PTK6 by PSPC1-CT-131, but not Mut-NLS (Fig. 6i).

We next determined the antitumor roles of PSPC1-CT131 in the HCC mouse model. First, parental and PSPC1-CT131-expressing Mahlavu cells were injected into nude mice to establish a xenograft model (Fig. 7a). After 4–5 weeks of tumor growth, mice injected with PSPC1-CT131-expressing cells showed a significantly decrease in tumor volume compared to parental cells. To explore the anti-metastatic potential with the pharmacological inhibitor of PSPC1-CT131 aiming to target both PSPC1 and PTK6, we evaluated whether the PSPC1-CT131 plasmid packaged and delivered in vivo with jetPEI®, an FDA approved in vivo transfection reagent, could suppress the lung metastasis induced by the systemic administration of PSPC1-expressing Mahlavu cells with a therapeutic procedure (Fig. 7b). Remarkably, treatment with PSPC1-CT131 reduced lung metastasis as shown by BLI images (Fig. 7c), the quantified BLI measuring metastatic tumor size (Fig. 7d) and the number of tumor nodules (Fig. 7e). Notably, treatment with PSPC1-CT131 also prolonged survival of mice compared to the untreated control mice (Fig. 7f). Furthermore, IHC analysis of metastatic lung tumor nodules indicated that the expression of PSPC1, PTK6 and β-catenin was significantly reduced in PSPC1-CT131-treated mice compared to untreated mice (Fig. 7g). In summary, our data indicated that PSPC1-CT131 interacted with PSPC1 and PTK6 in the nucleus to suppress synergistic oncogenic PSPC1/PTK6 signaling and abolished Wnt3a and TGF-β1 autocrine signaling to abrogate tumor progression (Fig. 8a–c for the hypothetical model and Supplementary Fig. 7h).

## Discussion

Although metastatic mechanisms via nucleocytoplasmic shuttling of cancer proteins such as PTK6 and β-catenin have been proposed in the past few decades[18,19,39], the cellular determinant that switches the oncogenic subcellular translocation of PTK6/β-catenin facilitating tumor cell metastasis remains poorly understood. Our results revealed that PSPC1 acts as the contextual determinant of the subcellular translocations of PTK6 and β-catenin and of their synergistic tumorigenic signaling by way of dynamic interchangeable interactions involving PSPC1/PTK6/β-catenin. We also demonstrated that PSPC1-CT131 is a dual pharmacologically

inhibitor targeting oncogenic PSPC1 and PTK6 and might be an advanced avenue for exploring clinical interventions to prolong survival of cancer patients.

Previous studies indicated that nuclear tyrosine kinase PTK6 might function as a tumor suppressor to phosphorylate RNA-binding proteins, such as Sam68, to restrain their proliferative and anti-apoptotic functions in divergent cancer cells and tissues[10,11,14,15,40]. The role of PSPC1 as the contextual determinant of the subcellular localization of cancer-related proteins via nucleocytoplasmic shuttling provides supporting evidence that PSPC1 is involved in switching PTK6 function from a tumor suppressor in the nucleus to an oncogene in the cytoplasm[38]. In addition, the concordant high expression of nuclear PTK6 and p-Y523-PSPC1 in lower grade HCC is associated with better survival of HCC patients, indicating that high expression of p-Y523-PSPC1 is a favorable prognostic biomarker for HCC patients and reflects the tumor suppressive function of PTK6 at early stages of HCC. In contrast, a decrease in the expression of p-Y523-PSPC1 is a warning of the synergistic oncogenic activation of PSPC1, nuclear β-catenin and cytoplasmic PTK6 that facilitates tumor progression in HCC patients.

Our finding of PSPC1 as a substrate that sequesters PTK6 tyrosine kinase in the nucleus provides an additional example for the crucial role of tyrosine phosphorylation of PTK6 nuclear substrates in tumor suppression. PSPC1, similar to other PTK6 nuclear substrates, such as transcription or splicing factors/cofactors Sam68, PSF, β-catenin, and parafibromin, functions in activating specific set of target genes to exert its biological functions[7,14,25,29]. For instance, PTK6 interacts and phosphorylates PSF to induce PSF cytoplasmic relocalization and cell cycle arrest in the breast cancer cell line BT-20 via the c-terminal tyrosine residue of PSF and the SH3 domain of PTK6[12]. With PSPC1-CT131 expressed in the nucleus, we suggested that PSPC1-CT131 could bind to the SH3 domain of PTK6 in the nucleus but not PSF in the cytosol, which may serve as an additional mechanism to suppress tumor progression. Moreover, tyrosine-dephosphorylated parafibromin cooperates with several oncogenic TFs/complexes, such as Gli1, Notch/β-catenin, and TAZ/TEAD/β-catenin/TCF, whereas it loses these functions upon tyrosine phosphorylation by PTK6[29,41]. Since upregulated-PSPC1 switches its binding partner from PTK6 to β-catenin during HCC tumor progression, future examination of the participation of PSPC1 in and the inhibitory effects of PSPC1-CT131 on Hippo oncogenic pathway is warranted for exploring potential therapeutic intervention to prolong HCC patient survival.

Nucleocytoplasmic shuttling is an ancient and complicated process that requires the participation and interaction of nuclear import and export proteins to sustain the cellular compartmentation needed to separate biological events and

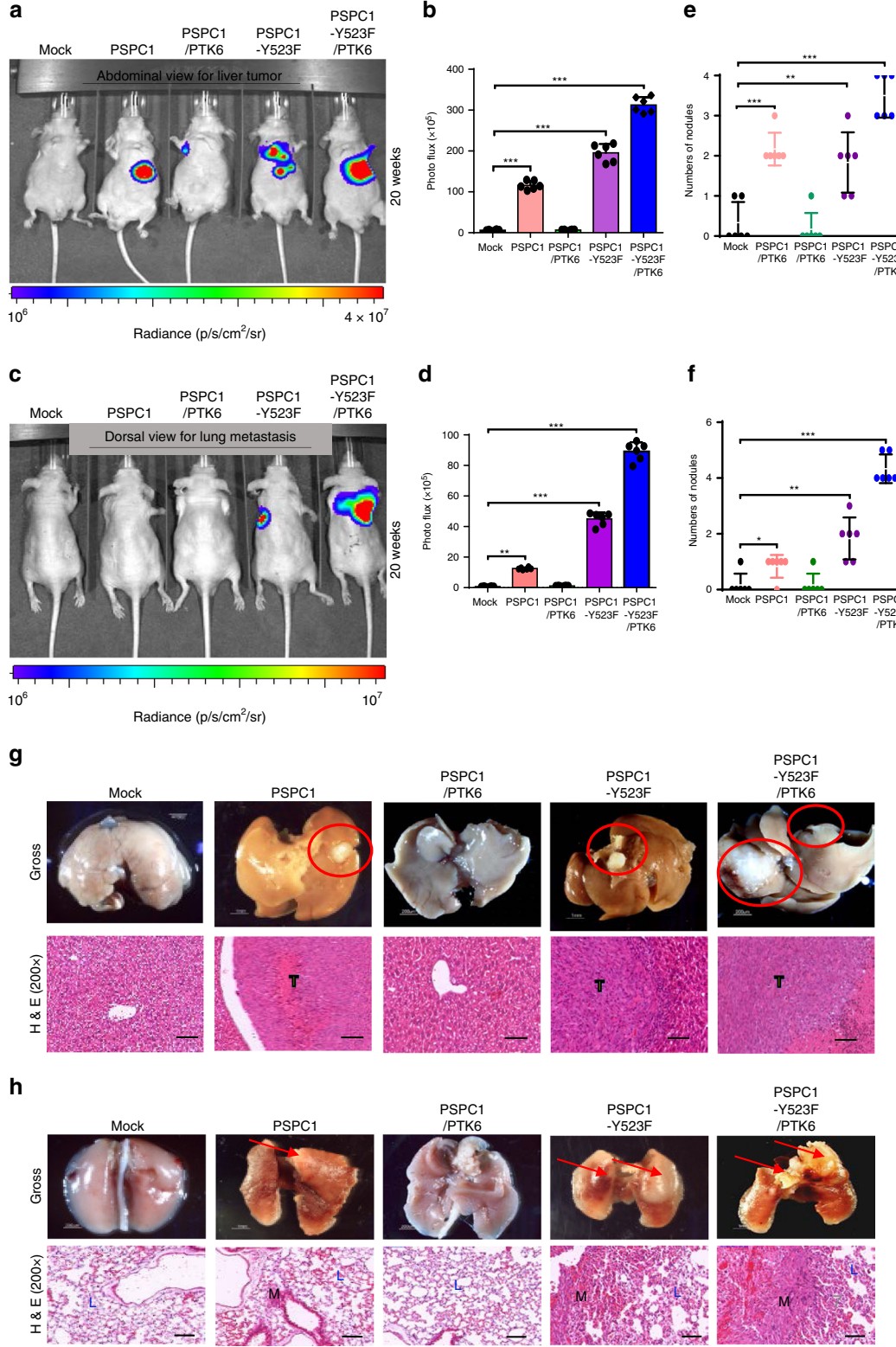

**Fig. 4 The PSPC1/PTK6/β-catenin axis is critical for tumor growth and metastasis in HCC. a–f** Effects on tumorigenesis (abdominal view for measuring liver tumors) (**a**, **b**) and metastasis (dorsal view for measuring lung metastatic tumors) (**c**, **d**) of the PSPC1/PTK6 interaction and PSPC1-Y523F mutant in orthotopic mouse HCC model generated by SK-hep1 transfectants bearing luciferase expression constructs. The effects were measured by representative bioluminescence images (**a**, **c**), intensity of photon flux representative of tumor size at 20 weeks (**b**, **d**), and numbers of nodules in tumors at 20 weeks (**e**, **f**). Data represent the mean ± SEM ($n = 6$/group). The PSPC1-Y523F mutant further enhanced tumorigenesis and metastasis in comparison with PSPC1 transfectants. All data statistics were based on *$p < 0.05$, **$p < 0.01$, and ***$p < 0.001$ calculated by one-way ANOVA with Bartlett's test.

**g**, **h** Representative images of livers and lungs showing tumorigenesis and metastasis (top panels) and hematoxylin and eosin-stained images of tumors from liver (**g**) or lung (**h**) isolated from representative mice, respectively. Red circles indicate locations of primary tumor nodules; red arrows indicate lung metastasis nodules; T, tumor; L, lung; M, metastasis. The scale bar represents 100 μm. Source data are provided as a Source Data file.

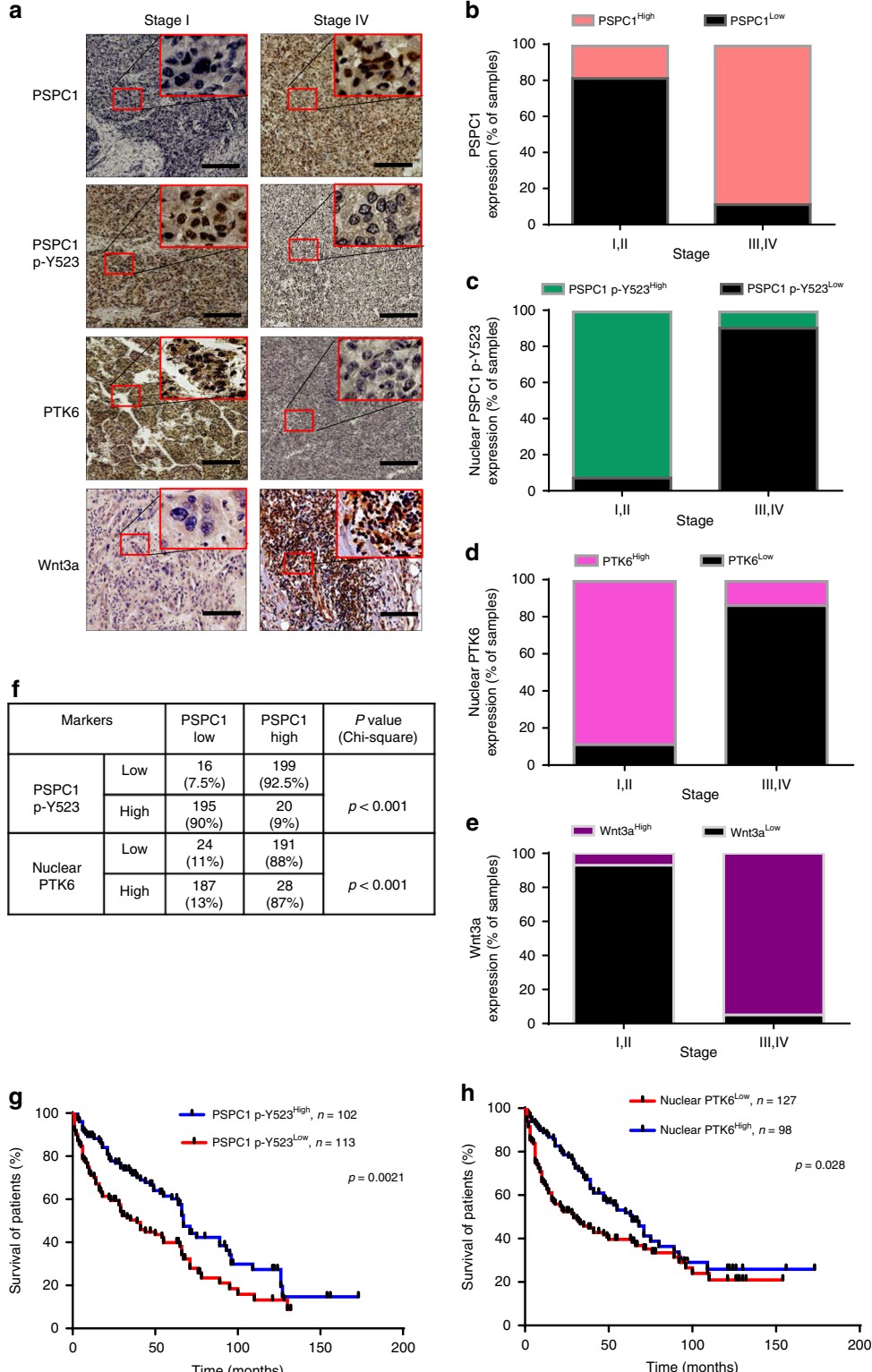

**Fig. 5 Phospho-Y523 PSPC1 is associated with better prognosis of HCC. a** Representative IHC staining results of PSPC1, PSPC1-pY523, PTK6, and Wnt3a expression in stage I (left panels) and stage IV (right panels) human HCC tissues. A large red tangle shows the enlargement areas is placed on the top right corner of each IHC image. The scale bar represents 50 μm. **b**–**e** Expression percentage of PSPC1 (red color for higher expression), phosphorylated Y523 PSPC1 in the nucleus (green color for higher expression), nuclear PTK6 (pink color for higher expression) and Wnt3a (purple color for higher expression) staining in early (I and II) and late stage (III and IV) tumors in human HCC tissues. Data are shown as the mean percentage of samples, $n = 215$. **f** The relationship between PSPC1 expression and the expression of PSPC1 phospho-Y523 and nuclear PTK6 in human HCC tissues. **g**, **h** Kaplan–Meier plot analysis of the overall survival of 215 HCC patients based on low or high expression levels of PSPC1-pY523 (**g**) and nuclear PTK6 (**h**). The *p* value was determined by the Gehan–Breslow–Wilcoxon test. Source data are provided as a Source Data file.

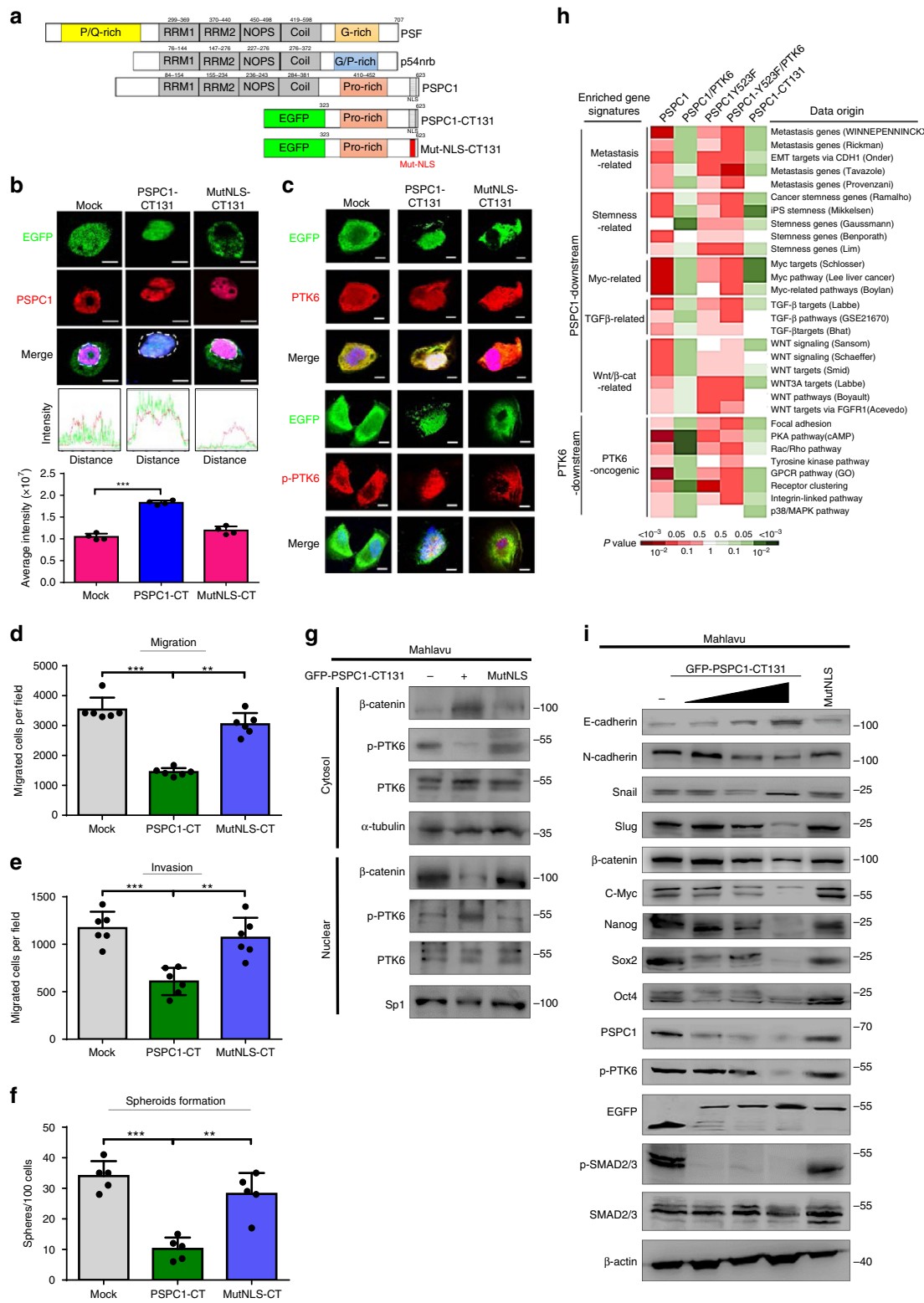

homeostasis for optimizing cell and tissue functions[39]. Aberrant subcellular localization of tumor suppressor genes and oncogenes has emerged as a hallmark of cancer with great potential to serve as therapeutic biomarkers for cancer intervention[42]. Strategies targeting different components in the nucleocytoplasmic transport process, including upstream regulatory signaling components, NLS and nuclear export signals (NES), and different interactions with transport proteins, including cargo proteins, importins, exportins, RanGTP, and nuclear pore complex proteins, are under intensive study for developing therapeutic inhibitors either in preclinical or clinical trials[43,44]. For example, Selinexor (KPT-330), a selective inhibitor of nuclear export (SINE) that targets CRM1 (chromosome region maintenance 1 protein, exportin 1 or XPO1), has been under intensive clinical trials either in monotherapy or in combined therapy with other therapeutic agents in diverse cancer types and has shown selective anticancer activity to cancer cells while sparing normal cells[24,45]. Molecular mechanisms of antineoplastic activity of SINE could be

**Fig. 6 The PSPC1-CT131 is a dual inhibitor of oncogenic PSPC1 and PTK6. a** A cartoon of the primary domain structures of aligned DBHS family proteins with PSPC1-CT131 and Mut-NLS-CT131 (nuclear localization sequence (NLS) mutation of PSPC1-CT-131). **b** Top: PSPC1-CT131, but not Mut-NLS-CT131, colocalized with PSPC1 in the nucleus in Mahlavu cells shown by IF images. Middle: the line graphics of colocalization of PSPC1 (red) and EGFP-PSPC1-CT131 (green). Bottom: summary of merged color intensities of EGFP, EGFP-PSPC1-CT131, and EGFP-Mut-NLS-CT131 (green) with PSPC1 (red) and DAPI (blue for DNA) expressed in Mahlavu cells. The merged color intensities were calculated based on areas marked with dashed circles and confocal immunofluorescence analysis of data representing the mean ± SEM ($n = 4$). The scale bar represents 10 μm. **c** PSPC1-CT131 but not Mut-NLS-CT131 sequestered the active form of p-PTK6 in the nucleus of Mahlavu cells demonstrated by IF analysis. Colors are active PTK6 (p-PTK6) and PTK6 in red, EGFP for PSPC1-CT131 and Mut-NLS-CT131 in green, and nuclei stained with DAPI in blue. The scale bar represents 10 μm. **d–f** Cell migration (**d**), invasion (**e**), and spheroid formation (**f**). Data represent the mean ± SEM ($n = 6$). All data statistics were based on *$p < 0.05$, **$p < 0.01$, and ***$p < 0.001$ calculated by one-way ANOVA with Brown–Forsythe test. **g** Expression of PSPC1-CT131 but not Mut-NLS-CT131 altered the subcellular localization of PTK6, p-PTK6, and β-catenin determined by Western blotting analysis in Mahlavu cells. **h** Comparison of statistical significance based on the expression intensity of PSPC1 and PTK6 downstream genes in gene set enrichment signatures after transcriptome analysis (GSE114856) of indicated proteins in SK-hep1 cells and PSPC1-CT131 in Mahlavu cells in heatmap displays. Red and green colors indicate upregulation and downregulation of matched gene sets, respectively. The deeper the color, the lower the $P$-value, as shown in the scale bar. **i** Western blotting analysis indicated that the expression of EGFP-PSPC1-CT131 induced the upregulation of E-cadherin but decreased the expression of N-cadherin, Snail, Slug, Nanog, Oct4, p-PTK6, β-catenin, C-myc, PSPC1, and p-Smad2/3 in a dose-dependent manner. Source data are provided as a Source Data file.

due to the inhibition of transporting activity of XPO1 resulting in nuclear accumulation of tumor suppressor proteins such as TP53, p27, p21, and others leading to tumor suppression[46,47].

Compared to other inhibitors that globally affect nucleocytoplasmic shuttling, PSPC1-CT131 selectively affected the subcellular localization switch to sequester PTK6 in the nucleus, suppress PSPC1-mediated tumor progression, sustain cytoplasmic expression of inactive β-catenin and abolish downstream autocrine signaling of TGF-β1 and Wnt3a, which act in concert to mediate tumor suppression. Our results revealed that PSPC1-CT131 is a different class of anticancer reagents in HCC that warrants future optimization for tumor suppressive applications and other cancers. Together, our results uncovered PSPC1-CT131 as a first in its class dual inhibitor targeting oncogenic PSPC1 and PTK6/β-catenin oncogenic subcellular translocation switch and might be a promising avenue for exploring clinical interventions to prolong the survival of cancer patients.

## Methods

**Antibodies**. The antibodies used in this study are listed in Supplementary Table 4.

**Cell culture, plasmids, and transfection**. HCC cell lines SK-hep1, Huh-7, SNU-387, Mahlavu and human embryonic kidney cell line 293T and 293FT cells obtained from ATCC and maintained in low passage culture as previous described[4]. SK-hep1 cells labeled with firefly luciferase were established by lentivirus infection. The full-length PSPC1 cDNA was cloned into pcDNA3.0-HA plasmid. The PSPC1-CT131 was further subcloned into pEGFP-C2 plasmid for confocal microscopy analysis. The QuikChange II Mutagenesis kit (Agilent) was used to generate all PSPC1 mutants in a pcDNA3.0-HA plasmid including PSPC1 CT131-NLS mutant (R409A) and Y523F mutant. We used the QuickChange II primer design website to generate primers for mutagenesis. For transfection experiments, jetPRIME transfection reagent (Polyplus) was used and stable expressing clones were derived with G418 or zeocin selection for in vivo experiments. The expression plasmids of Nanog, Oct4, Sox2, and the promoter constructs of Snail, Slug were purchased from Addgene. The Twist promoter plasmid is a gift kindly provided by Dr. L.H. Wang at National Health Research Institutes of Taiwan. The β-catenin del-N mutant plasmid and the dominant-negative TCF4 (TCF-DN) plasmid was kindly provided by Dr. Hsiu-Ming Shih at the Institute of Biomedical Sciences of Academia Sinica.

**Cell migration and invasion assays**. Cell migration assay was performed using Boyden chambers. For the invasion assay, each transwell was coated with matrigels (BD Biosciences). The upper insert contained $1 \times 10^4$ cells in 200 μl serum free medium was placed onto the lower chamber filled with 800 μl complete medium as chemo-attractant. After 24 h, cells were fixed with methanol for 10 min. The matrigels and un-migrated cells were removed by using cotton swabs. Subsequently, the chambers were stained with Giemsa stain (Merck) and migrated cells were counted in 100× microscopic fields (inverted microscope, Nikon).

**Enzyme-linked immunosorbent assay (ELISA)**. CM from each group was collected and stored at −80 °C prior to ELISA analysis. The concentration of WNT1, WNT3a was measured by Human WNT1 and WNT3a ELISA kit (AVIVA

SYSTEMS BIOLOGY) and total TGF-β1 was measured by Human TGF-beta1 Platinum ELISA kit (Ebioscience) according to the manufacturer's instructions.

**An orthotopic tumor model for hepatocellular carcinoma**. A 6- to 8-week-old male Swiss nu/nu mice were purchased from National Laboratory Animal Center, Taipei, Taiwan and were housed and maintained under specific pathogen-free conditions. All mouse experiments were conducted with approval from the Experimental Animal Committee, Academia Sinica. For orthotopic transplantation, cells were re-suspended ($1 \times 10^6$ cells/0.05 mL HBSS) with Matrigel (Corning, Coring NY, USA; Cat. no.: 354230) and injected into the liver. For lung metastasis and primary liver tumor growth, mice were sacrificed at week 20 after implantation. Tumors in liver tissues and metastatic lung were monitored by the bioluminescent imaging analysis using IVIS image system. Mice were sacrificed for examining nodules in lung metastasis and primary liver tumors.

**GST pulldown assay**. The cDNA fragments for PSPC1 were cloned into pGEX-4T-1 vector to generate glutathione S-transferase (GST) fusion proteins and assays were performed using recombinant GST-PSPC1 and PTK6 (Thermo-Fisher Scientific). For the GST pull-down assay, GST-tagged PSPC1 was incubated with recombinant PTK6 in GST pulldown buffer (100 mM Tris [pH 8.0], 1% NP40, 150 mM NaCl) overnight at 4 °C and then washed six times. The bound proteins were analyzed by Western blotting.

**Gene set enrichment analysis (GSEA)**. GSEA was performed on various gene signatures by comparing gene sets from MSigDB database or from published gene signatures. Gene sets with a false discovery rate (FDR) value <0.05 by comparing the enrichment score to enrichment results generated from 1000 random permutations were considered as statistical significance. All the original GSEA data was provided in Supplementary Table 2.

**Phospho-specific antibody against p-Tyr523-PSPC1**. Rabbit polyclonal antibody against phospho-Tyr523 of PSPC1 was raised against phosphopeptide: CGGNFEGPNKRRR(Yp) synthesized by LTK Biolaboratories (Touyuan, Taiwan). Antibodies were affinity-purified by binding to a phosphopeptide column. The eluted antibodies were purified by passing through the un-phosphorylated peptide column to remove antibodies that cross-react with un-phosphorylated epitopes.

**Immunoprecipitation**. Cells were lysed with radioimmunoprecipitation assay (RIPA) buffer (50 mM Tris-Cl pH8.0, 150 mM NaCl and 1% NP-40). The primary antibody or control IgG with Protein A/G Sepharose Beads (GE) was added into the lysates and incubated at 4 °C for overnight. The beads with precipitated proteins were collected and washed by RIPA buffer before immunoblotting analysis.

**Immunohistochemistry**. After deparaffinization, the tissue sections were subjected to 10 mM citrate buffer (pH 6.0) by microwave treatment for 20 min for antigen retrieval. The samples were subsequently immersed in 3% $H_2O_2$ for 30 min to block endogenous peroxidase, and then incubated with anti-PSPC1, anti-PTK6 or anti-PSPC1 phospho-Y523 primary antibody diluted in blocking buffer at 4 °C overnight. The slides were processed using the SuperPicture™ Polymer Detection kit (Thermo-Fisher Scientific) according to the manufacturer's protocol, and counterstained using hematoxylin. Tissue arrays were purchased from SUPER BIO CHIPS (www.tissue-array.com, Seoul, Korea). All IHC results and the association with patient survival were examined and scored from 1 to 4 multiplied with percentage of stained cells (H-score) based on their expression intensity by two independent pathologists. High expression of particular protein staining was

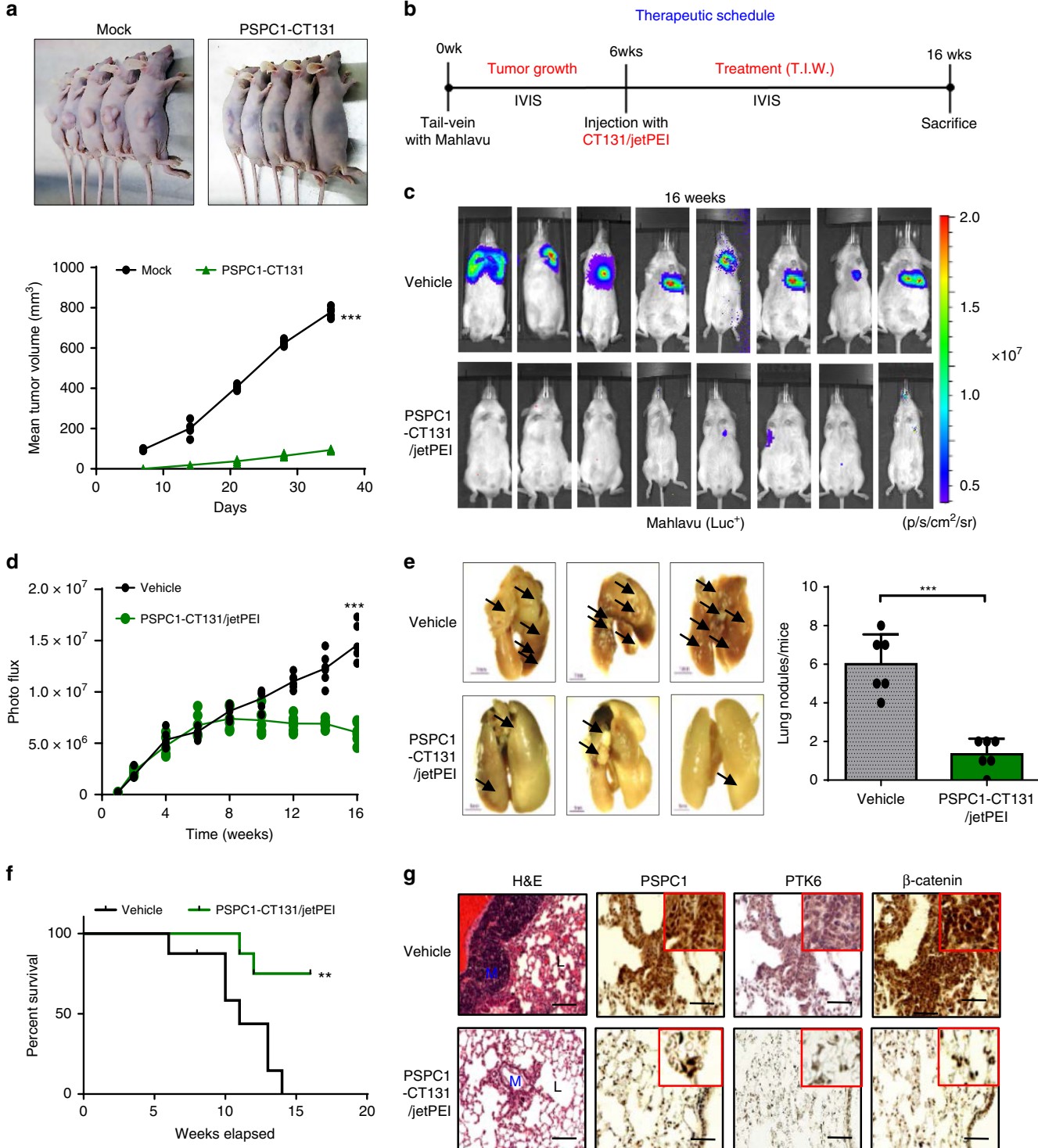

**Fig. 7 PSPC1-CT131 targeting of PSPC1 and PTK6 reduces metastasis in HCC. a** Representative images of nude mice (top) and the tumor growth curves (bottom) in nude mice injected with parental and PSPC1-CT131-overexpressing Mahlavu cells. Tumor volumes at 6 weeks post injection are presented as the mean ± SEM ($n = 5$/group). Data statistics were based on *$p < 0.05$, **$p < 0.01$, and ***$p < 0.001$ calculated by ordinary two-way ANOVA with post hoc Tukey's test compared to the mock control. **b** Time flowchart of the therapeutic schedule of tail vein injections of PSPC1-CT131/jetPEI® into mice carrying tail vein injected Mahlavu cells. **c** Representative bioluminescence images of experiments in (**b**). **d**–**f** Injection of PSPC1-CT131 plasmid packaged in vivo with jetPEI® not only reduced lung metastasis (**d**, **e**) but also prolonged mouse survival (**f**) in the tail vein injected lung metastasis model ($n = 8$/groups). Data statistics were based on *$p < 0.05$, **$p < 0.01$, and ***$p < 0.001$ calculated by Log-rank test compared to vehicle control. **g** Validated expressions of the PSPC1/PTK6/β-catenin axis by IHC staining of lung tumors after the administration of PSPC1-CT131 in a mouse lung metastasis model. Red squares on the top right corner of each IHC image stand for enlargement of stained tissues. The scale bar represents 100 μm. Source data are provided as a Source Data file.

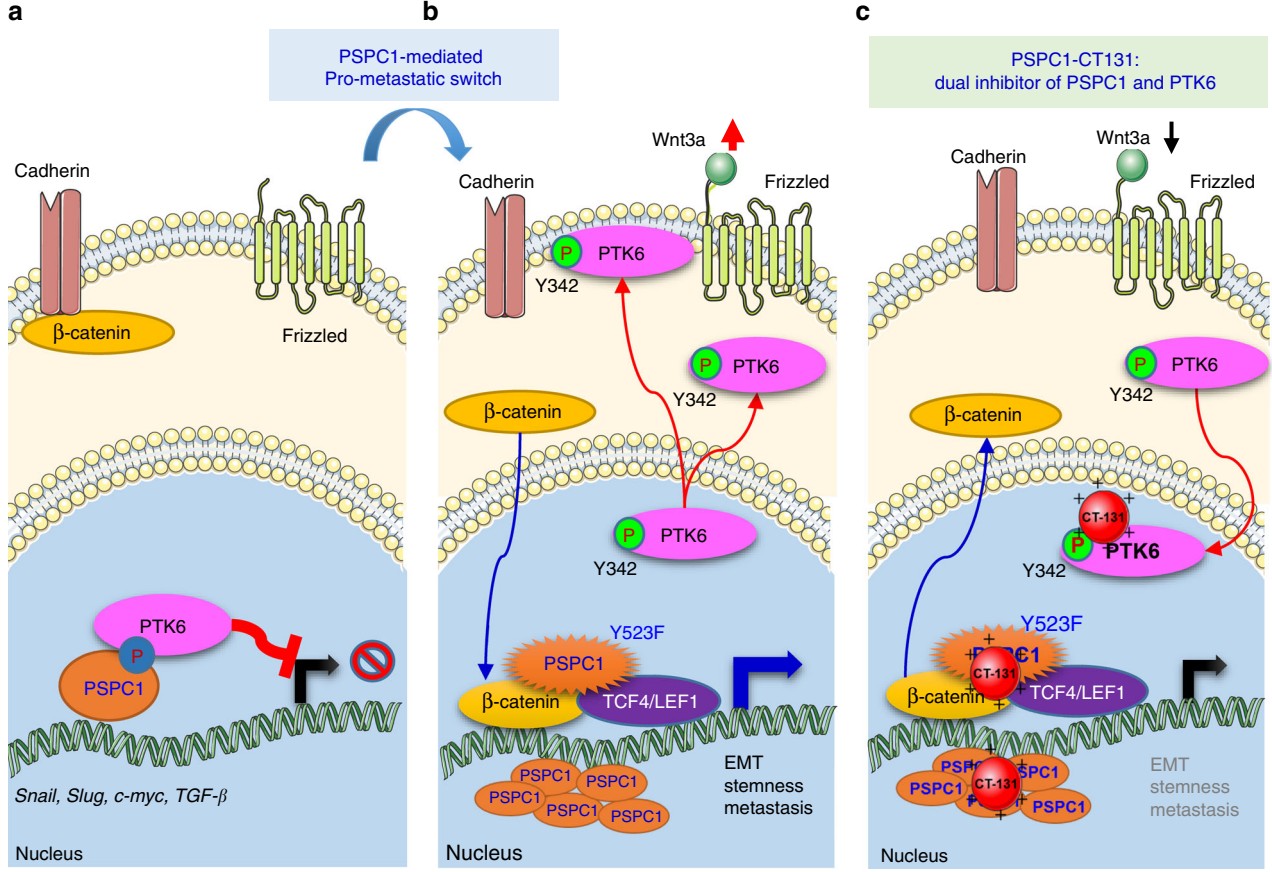

**Fig. 8 Hypothetical models of the PSPC1/PTK6 interaction modulating Wnt/β-catenin autocrine signaling and PSPC1-CT131 as a dual inhibitor of PSPC1 and PTK6 suppressing downstream oncogenic signaling. a** PTK6 suppressed PSPC1 oncogenic features by phosphorylation and interaction on Y523 of PSPC1 and was sequestered in the nucleus as a tumor suppressor. **b** PSPC1 upregulation or mutation of the PTK6 phosphorylation site PSPC1-Y523F augmented Wnt3a autocrine signaling to facilitate β-catenin nuclear and p-PTK6 cytoplasmic/membrane translocations to synergize the oncogenic signaling of both PSPC1 and PTK6, which facilitated EMT, stemness and metastasis. **c** Treatment of PSPC1-CT131 as a dual inhibitor of PSPC1 and PTK6 reduced the expression of both oncogenic PSPC1 and PTK6 to suppress HCC tumor progression. Overall, PSPC1-CT131, a dual inhibitor of PTK6 tyrosine kinase and PSPC1 pro-metastatic master activator, is a potent anticancer agent that prolongs the survival of mice in an HCC mouse model.

defined with the intensity of discriminatory score excess 200. All the original IHC data was provided in Supplementary Table 3. The use of human cancer tissue arrays from commercial sources was approved by the Human Subject Research Ethics Committee/IRB of Academia Sinica. For preparing IHC blocks from cell lines, $1 \times 10^6$ cells were re-suspended in 2% ultra-low gelling temperature agarose (Sigma) followed by placed on ice until solidified of agarose blocks. The solid agar was further fixed with 4% paraformaldehyde and then for embedding.

**Immunofluorescence microscopy.** Cells were plated onto glass coverslips, fixed with 4% paraformaldehyde, permeabilized by cold methanol and 0.1% Triton and stained with primary antibody and then with the corresponding Alexa Fluor-488 or Alexa Fluor-594-conjugated secondary antibody (Thermo-Fisher Scientific). The antibody-labeled cells on coverslips stained with 4′, 6-diamidino-161 2-phenylindole (DAPI) were then mounted by ProLong™ antifade mountant with DAPI (Thermo-Fisher Scientific) for analysis. Images were obtained by confocal laser-scanning microscopy. Serum-starved and confluent cultures of SK-hep1 cells were used for detection of PSPC1, PTK6, γ-catenin and N-cadherin expression and Mahlavu cells were used for detection of EGFP, E-cadherin and N-cadherin expression.

**In vivo metastasis assay.** We followed the guidelines of the Institutional Animal Care for all the animal experiments with approval by the Animal Committee of Academia Sinica. For systemic metastasis assay, the lung metastasis model was established by tail-vein injection of Mahlavu ($1.0 \times 10^6$/μL cells) cells into each 6- to 8-week-old male NOD/SCID mice in groups of eight mice. Mice were purchased from BioLASCO Taiwan Co., Ltd. Lung metastatic signals were detected by using the IVIS system (Xenogen Corp.) with the excitation and emission wavelength at 570 and 620 nm. The mice were sacrificed at 16–20 weeks after injection and the lungs were removed and fixed in 4% paraformaldehyde. The detectable tumor nodules on the surface of whole lung were counted for metastatic index.

Histological staining was used to further confirm the presence of PSPC1/PTK6/Wnt signaling in lung metastases.

**Luciferase reporter assays.** Luciferase activities of firefly and renilla were measured by Dual-Glo® Luciferase Assay System (Promega) based on the manufacturer's instructions.

**MS analyses.** The IP of PSPC1 protein complexes were fractionated by SDS-PAGE, followed by instant Coomassie blue staining (Expedeon). The Coomassie blue-staining gel bands were manually sliced, destained and digested with sequencing-grade trypsin (Promega). The peptide mass and peptide fragment mass were measured by LC–ESI/MS/MS or 2D-LC-ESI/MS/MS followed by identifying the matched proteins to the NCBI and SwissProt databases. The original results were provided in Supplementary Table 1.

**Protein 3D structure prediction.** We retrieved the sequences of PSPC1 and PTK6 from NCBI database. Since there is no structure solved for both proteins, we performed the protein structure prediction by using I-TASSER (http://zhanglab.ccmb. med.umich.edu/I-TASSER/). From the top models predicted by I-TASSER, we chose the structures with best scores for further analysis. The protein–protein docking predictions were performed by using ClusPro (http://cluspro.bu.edu/help.php).

**Therapeutic administration of PSPC1-CT131 plasmid.** Solutions of PSPC1-CT131 and vehicle negative control were each diluted with in vivo-jetPEI solution (catalog no. 201-50G, Polyplus-transfection) containing 10% (wt/vol) glucose at a ratio recommended by the manufacturer. All solutions were mixed by vortexing for 10 s and incubated for at least 15 min at 37 °C before injection. Each mouse received 200 μl glucose and saline and PSPC1-CT131 plasmid (100 μl of PSPC1-CT131 plasmid solution plus 100 μl of saline through tail vein injection consecutively for 3 days and 3 additional injections were performed once a week for

the following 8 weeks. Two additional groups of control animals were included: one consisting of untreated animals and the other of animals receiving a mixture of in vivo-jetPEI® solution containing 10% (wt/vol) glucose without added PSPC1-CT131 plasmid.

**RNA sequencing (RNA-Seq) and data analysis.** Total RNAs isolated from Mock control cells, PSPC1-overexpressing cells, PSPC1/PTK6-overexpressing cells, PSPC1-Y523F-overexpressing cells, PSPC1-Y523F/PTK6-overexpressing cells in SK-hep1 cells. Mock control cells, PSPC1-CT131-overexpressing cells, and PSPC1-mutNLS-overexpressing cells in Mahlavu cells were extracted by TRIzol™ reagent (Invitrogen). RNA quality was examined by spectrophotometry, agarose gel electrophoresis (18S and 28S rRNA ratio) and Agilent Technologies 2100 Bio-analyzer with an RNA integrity number value greater than 8. After rRNA depletion (Epicenter), RNA fragmentation and library preparation (Illumina), the constructed libraries were then performed 150 bp paired-end sequencing by an Illumina NovaSeq6000 sequencer at Tools (Taiwan). Expression analysis was performed and analyzed by Novogene (Tools, Taiwan). Briefly, reads were aligned to the hg18 genome build and FPKMs were quantile normalized across all samples.

**RNA preparation and quantitative reversed transcription PCR.** Total cellular RNA was extracted using TRIzol™ reagent (Invitrogen) for RT-PCR. RNA was used for reverse transcription with Quant II fast reverse transcriptase kit (Tools, Taiwan). Quantitative RT-PCR was performed by using the SYBR Green Master Mix (Applied Biosystems) according to the manufacturer's protocols. The specific primers used in RT-qPCR are listed in Supplementary Table 5.

**Small hairpin RNA and lentiviral infections to cells.** The small hairpin RNAs (shRNAs) for PTK6 were obtained from the TRC library: TRCN0000021552 (shRNA52) and TRCN0000199853 (shRNA53) as shRNA for PTK6 from the National RNAi Core Facility Platform of Academia Sinica. pLKO.1 with shRNA, pMD.G and pCMV-ΔR8.91 were introduced into HEK293FT cells for lentiviral packaging. The viral supernatants were collected for infection of HCC cancer cell lines. Control vector expressing shRNA against LacZ (pLKO.1-shLacZ) was used as a negative control.

**Subcellular fractionation and Western blotting analysis.** Nuclear and cytoplasmic fractions were prepared using the Nuclear and Cytoplasmic Extraction reagents (Thermo-Fisher Scientific) as manufacturer's instruction. Total cellular proteins were extracted using RIPA lysis buffer and then quantified protein concentration by BCA protein assay kit (Pierce). The protein lysates were separated on SDS-PAGE, electro-blotted onto PVDF membranes (Millipore), probed with primary antibody followed by HRP-conjugated secondary antibody, and then detected by enhanced chemiluminescence.

**Spheroid formation assays.** Totally, 1000 cells were suspended in DMEM/F12 medium containing 20 ng/ml EGF, 20 ng/ml basic FGF and B27 supplements (Thermo-Fisher Scientific). Cells with limiting dilutions were cultured in 12-well plates for 2 weeks. Spheroids larger than 20 μm were then counted for spheroid-forming index.

**Side population detection.** Cells were harvested and adjusted with fresh medium to the cell density of $1 \times 10^6$ cells per ml. Aliquots thereof were put aside for control purpose, and either verapamil (20–100 μM) (Catalog number V4629), or reserpine (20–100 μM) (Catalog no. 83580) was added. Verapamil and reserpine are known to block several ABC drug transporters (Sigma-Aldrich). Cells are distinguished from debris on the flow-cytometric profile based on the forward scatter and side scatter (SSC). Cell doublets and aggregates are gated out based on their properties displayed on the SSC area versus height dot plot. Side population (SP) cells are recognized as a dim tail extending toward the lower "Hoechst Blue" signal. The gating tree indicates the sequential procedure applied to select out the final population for SP discrimination and the percentage of cells (gated events) resulting from each gating step. The gating strategy are represented as shown in Supplementary Fig. 8. Cells were filtered using 35 μm filter round-bottom FACS tubes (BD Biosciences) immediately before data acquisition on either an LSR II SORP or Fortessa (BD) and data analyzed using FlowJo software (Tree Star, Inc.).

**Tumor xenograft.** We followed the guidelines of the Institutional Animal Care for all the animal experiments with approval by the Animal Committee of Academia Sinica. Six-week-old female BALB/c nude mice were purchased from National Laboratory Animal Center (Taipei, Taiwan) and were maintained under specific pathogen-free conditions. In order to establish a subcutaneous xenograft model, PSPC1-CT131 and control groups in Mahlavu cell were suspended at a concentration of $1 \times 10^7$ cells/ml mixed with Matrigel for subcutaneous injection. Tumor incidence were monitored at 7 weeks after injection ($n = 5$). Statistical test is two-way ANOVA with post hoc Tukey's test. The data represent the mean ± SEM.

**Statistical and Kaplan–Meier survival analysis.** Data were expressed as the mean ± SEM. All statistical analyses were conducted using GraphPad Prism 7.0 statistical (http://www.graphpad.com/scientific-software/prism) software. Statistical significance was set at *$p < 0.05$, **$p < 0.001$ and ***$p < 0.0001$ by two-tailed Student's $t$ test and one-way ANOVA. Survival durations were analyzed using the Kaplan–Meier method and compared by the log-rank test in the patient groups.

**Reporting summary.** Further information on research design is available in the Nature Research Reporting Summary linked to this article.

## Data availability
The data from the Gene Expression Omnibus (GEO) database analyzed for this study is GSE114856. All the other data supporting the findings of this study are available within the article and its Supplementary Information files. The source data underlying Figs. 1a–j, 2a–j, 3a–h, 4b–f, 5b–h, 6b–i, 7a, d, e, f and Supplementary Figs. 1c–h, 3c, 4c–g, 6b, and 7e–h are provided in the Source Data file.

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

## Acknowledgements
We thank Common Equipment Core of IBMS and Academia Sinica including micro-scopy, DNA sequencing, SPF animal facility (AS-CFII-108-103), Proteomics Core Facility, DNA sequencing (AS-CFII-108-115) and Flow Cytometry (AS-CFII-108-113) for supporting our experiments. We thank BIOTOOLS CO., LTD. for RNA-seq services. Our works are supported by grants of Taiwan from the Academia Sinica and Ministry of Science and Technology (MOST) [106-0210-01-15-02] and from MOST [107-2321-B-001-025] and [104-2320-B-001-009-MY3].

## Author contributions
Y.D.L. designed and performed the experiments, analyzed and interpreted the data, and participated in writing the paper. H.Y.C., E.C.H., R.S., H.W.Y., Y.C.L., J.W.C., and C.Y.W. performed the experiments; C.M.H. and J.H.S. performed the bioinformatics analysis; Y.D.L., H.Y.C., R.H.C., and Y.S.J. wrote the paper and were involved in the discussion of the results.

## Competing interests
The authors declare no competing interests.
