## [Peer Review File · Nature Communications]

Reviewers' comments:

Reviewer #1 (Remarks to the Author):

Paraspeckle component 1 (PSPC1) is shown to associate with PTK6 in HCC cell lines. Expression of PSPC1 Y523F abolishes interaction. Overall it is not clear if PSPC1 and its phosphorylation is playing any direct role or if its overexpression is simply modulating PTK6 activity. Brauer, Zheng et al. 2010 showed that targeting active PTK6 to the nucleus is growth inhibiting. Here, overexpression of a nuclear C-terminal proline rich fragment of PSPC1 (CT-131) that presumably binds the PTK6 SH3 domain and relieves PTK6 autoinhibition in the nucleus, has therapeutic activity. This would allow PTK6 to phosphorylate other PTK6 nuclear targets such as parafibromin (Hatakeyama and colleagues, 2017 and 2018) or even beta-catenin (Palka-Hamblin et al. 2010). It is interesting that CT-131 has therapeutic activities but its mechanism of action is not clear.

Comments:

There is no discussion of recent papers showing that parafibromin, a beta-catenin regulator, is a PTK6 substrate, and how this might impact the interpretation of the studies (Tang et al., 2018 and Kikuchi et al., 2017).

HGF is used "for induction of cancerous EMT microenvironment," but discussion of HGF activation of PTK6 (BRK) and cell migration through MET is not included (Lange and colleagues, 2010, 2012, 2013). In 3a HGF treatment leads to increased PTK6 in the cytoplasm; is it active, PY342 positive? Membrane fractions are not shown. Is HGF activating PTK6 signaling at the plasma membrane to promote oncogenesis?

Several reports indicate oncogenic roles for active PTK6 at the plasma membrane. Zheng et al. 2012 showed that P-PTK6 at the plasma membrane induces cell shape changes shown in Fig. 2B. Here, immunofluorescence images of active PTK6 are not shown in any panel and membrane fractions are not included in the immunoblotting studies.

Fig. 1a: You cannot definitively identify proteins on a Coomassie stained gel without immunoblotting.

Fig. 1g. How does EMT marker expression look with overexpression of WT PSPC1 alone?

Fig 2a. This panel does not demonstrate that PTK6 phosphorylates Y523 as indicated in the text. It shows that PSPC1 Y523 is required for PSPC1-PTK6 association. The impact of the Y523F mutation on PTK6 mediated PSPC1 phosphorylation should be shown. Supplemental 6c shows that PSPC1 PY523 associates with PTK6, not that PTK6 phosphorylates PSPC1 at this residue.

Fig. 2b. Are these really spheroids and not clusters? The fluorescent cells do not appear to be part of the spheroids as indicated in the legend. It would be interesting to include staining for PSPC1, PTK6, and P-PTK6 in Fig. 2b.

In Fig. 6f, CT131 increases activation of nuclear PTK6 but not cytoplasmic PTK6 and there does not appear to be a shift in total PTK6. Immunofluorescence staining for active P-PTK6 and total PTK6 should be shown.

There are several technical issues. Problems with the language are present throughout: a few examples, "A prometastatic activator upregulated-PSPC1 in cancers might emerge as the lethal metastasis," "Indigenous knockout," "we performed divergent PSPC1/PTK6 constructs," etc. Typos are also present throughout such as r-catenin (γ -catenin). Inclusion of size markers on the westerns is recommended. Quantitation of protein expression changes highlighted in the westerns would be helpful. Figures: 1a, d could be shown in supplemental data. TOP/FOP Flash constructs in Fig. 3e and f are not required.

Reviewer #2 (Remarks to the Author):

Lang et al. reported interesting aspects of PSPC1 functions. The authors first identified PTK6, which has tumor suppressive functions, as one of the major PSPC1-interacting proteins. They examined the importance of this interaction in cancer by using cell-based and mouse cancer models as well as cancer clinical data. Then they have shown regulatory roles of PSPC1 in the PTK6/beta-catenin axis. However, the drawback of this manuscript is to contain numerous grammatical errors and typos. In addition, one of the major conclusions is not fully supported by the data shown in the manuscript as described below.

Major points

1. The authors should consider consulting a language editing service. There are numerous grammatical errors, typos, and uncommon usage of terms in the manuscript. Thus, there are many ambiguous sentences, which often disturb reading. Even in the abstract, multiple sentences are not clear, and thus it would be possible not to convey the authors' thoughts to readers.
2. The authors claim that PSPC1-CT131 is a dual inhibitor targeting oncogenic PTK6 and PTK6, and the interaction between the CT-131 and endogenous PTK6 is shown in the model of Figure 8. But, no biochemical data for this interaction is shown in the manuscript. Previous studies have shown that PTK6 dimerize and oligomerize with DBHS family proteins via the NOPS domain and the coiled-coil domain, respectively, but C-terminal domain is not reported for contributing the interaction. The PSPC1-CT131 alone actually binds PTK6 (and Y523F mutant)?
3. Related to the above point, how does PSPC1-CT131 inhibit endogenous PTK6 functions? In Figure 6h, PTK6 expression was reduced by expression of EGFP-PSPC1-CT131. What is an interpretation for this result? Is this possible to be related to the mechanism of PTK6 inhibition by the CT-131?

Minor points

1. Fig. 1c: The labels of immunoblotting should be corrected to HA or FLAG as used in Figure S1b and S1c.
2. In Figure S2a, in vitro kinase assay using PTK6 and PTK6 WT is performed. The PTK6-Y523F mutant is actually not phosphorylated in this assay?
3. In middle cartoon of Figure 8, cytoplasmic PTK6 should be phosphorylated according to the data shown in Figure 3c?
4. Methods: In "Antibodies" section, information of cell lines and plasmids are included. Please fix this point.
5. Methods: Transfection methods are described in "PSPC1 site-directed mutagenesis" section. The transfection methods should be separated or combined with other sections.
6. In Figure 5b & c and Figure 6 c & d, no labels are found on the x-axis. I guess that these are same as in Figure 5d and Figure 6e, respectively. But, I think that this labeling is uncommon. Please fix this point.

Reviewer #3 (Remarks to the Author):

General comment: PTK6 kinase plays an opposite role in tumor progression which is dependent on its subcellular localization and transportation between nucleus and cytoplasm. PTK6 protein, a substrate of PTK6 kinase closed to splicing speckles and recently has been reported by Dr. Jou's group to regulate TGF- β signal pathway for pro-metastatic switches of cancer cells. In this study, Lang et al. showed that PTK6-Y523F cause oncogenic subcellular translocations of cytoplasmic PTK6 and nuclear β -catenin as well as Wnt3a autocrine loop for promoting liver cancer progression. PTK6-CT131 which is the proline-rich C-terminal domain of PTK6 containing NLS (nuclear localization signal), interacts with PTK6 in nucleus to inhibit PTK6/PTK6 oncogenic functions. The study is interesting and provides mechanistic insights for understanding the role of subcellular localization of PTK6/PTK6/ β -catenin

in regulating liver cancer progression. However, some key experiments supporting individual claims may not be sufficient to reveal causal and concrete regulatory mechanisms. Several issues should be clarified.

Comment 1: The SH2 domain interacts with phosphorylated tyrosine residues and SH3 domain binds to proline-rich region (Yeatman T. J., 2004). Does proline-rich region of PSPC1 directly bind to SH3 domain of PTK6 (Supplementary Figure 1b, c)?

Comment 2: The author claims that PSPC1 upregulation and PSPC1-Y523F released PTK6 nuclear sequestration; however, the existing evidence of this study for demonstrating the above results are not enough. In particular, the interaction between PSPC1 and PTK6 should be confirmed in different cellular fractions and the immunofluorescent images which indicate PSPC1-Y523F instead of PSPC1-wild type released PTK6 nuclear sequestration should be provided.

Comment 3: For demonstrating the interaction between PSPC1 and PTK6 and their subcellular translocation, almost experiments in this study were performed by ectopic gene expression system. To obtain more convincing evidence that PSPC1 interacts with PTK6 naturally, endogenous gene expression experiments should be performed.

Comment 4: Which kinase(s) phosphorylates PTK6?

Comment 5: In Supplementary Figure 5a IHC analysis, the expression of PTK6 in PSPC1-Y523F/PTK6 is lower than PSPC1/PTK6 group. How does mutant PSPC1 regulate PTK6 expression?

Comment 6: Supplementary Figure 7c, for clearly confirming the localization and expression of EGFP, EGFP images should be shown as an independent panel. Moreover, both N-cadherin and E-cadherin are cell-cell adhesion proteins, their localization should be at the cell-cell contacts.

Comment 7: The evidence for supporting PTK6/PSPC1 to regulate Wnt3a autocrine may not be sufficient. Detection of Wnt3a should in animal experiments and clinical samples is recommended.

Minor points:

1. For Figure 1h, 1k, 2d, 2f and 6d, the labeling of y-axis should be corrected to "Invasive cells per field".
2. In Supplementary Figure 1b, the author established a GFP-PSPC1-CT construct to confirm that Proline-rich C-terminal domain is a major binding site on PSPC1 for PTK6 binding. However, the full-length of PSPC1 is cloned in a HA vector. Both PSPC1-CT and full-length of PSPC1 should be individually cloned in the same vector.
3. For Figure 3b immunoprecipitation assay, the input of β -catenin, PSPC1 and PTK6 as well as internal control such as β -actin should be shown.
4. For Supplementary 4a, the groups of cell morphology in 2D culture are not labeled.
5. The spheroid images of Figure 2b are not clear.

Point-by-point response to the reviewer's comments:

Reviewers' comments:

All of the responses are underlined.

Reviewer #1 (Remarks to the Author):

Paraspeckle component 1 (PSPC1) is shown to associate with PTK6 in HCC cell lines. Expression of PSPC1 Y523F abolishes interaction. Overall it is not clear if PSPC1 and its phosphorylation is playing any direct role or if its overexpression is simply modulating PTK6 activity. Brauer, Zheng et al. 2010 showed that targeting active PTK6 to the nucleus is growth inhibiting. Here, overexpression of a nuclear C-terminal proline rich fragment of PSPC1 (CT-131) that presumably binds the PTK6 SH3 domain and relieves PTK6 autoinhibition in the nucleus, has therapeutic activity. This would allow PTK6 to phosphorylate other PTK6 nuclear targets such as parafibromin (Hatakeyama and colleagues, 2017 and 2018) or even beta-catenin (Palka-Hamblin et al. 2010). It is interesting that CT-131 has therapeutic activities but its mechanism of action is not clear.

Comments:

1. There is no discussion of recent papers showing that parafibromin, a beta-catenin regulator, is a PTK6 substrate, and how this might impact the interpretation of the studies (Tang et al., 2018 and Kikuchi et al., 2017).

Responses:

As suggested, we added a paragraph and four references in the “Discussion” section at **Page 23**. As following:

“Our study of PSPC1 as a substrate to sequester PTK6 tyrosine kinase in nucleus provides additional supports for the crucial role of nuclear tyrosine phosphorylation in modulating selective cofactors such as Sam68, β -catenin, PSPC1 and parafibromin of transcription factor complexes for activation of specific target genes to pursue the designated biological functions¹⁻⁴. For instance, parafibromin is a nuclear substrate of tyrosine phosphorylation by PTK6 and then de-phosphorylation by SHP2 to cooperate with different transcription factor

complexes such as phospho- parafibromin/YAP/TEAD and de-phospho- parafibromin/TAZ/TEAD/ β -catenin/TCF respectively for activation of different target genes^{4,5}”

2. HGF is used “for induction of cancerous EMT microenvironment,” but discussion of HGF activation of PTK6 (BRK) and cell migration through MET is not included (Lange and colleagues, 2010, 2012, 2013).

Responses:

Thank you for suggestions. We cited these papers and described in “Results” section at **Page 12**.

“We selected hepatocyte growth factor (HGF)/c-Met signaling as a cell model to potentiate EMT and stemness in Huh-7 cells^{2,6,7}.”

Moreover, we illustrated HGF-induced c-Met signaling by transcriptome analysis (**GSE114856, as shown in Supplementary Fig. 4c, d and Supplementary Table 2**) and suggested c-Met is the kinase to phosphorylate PTK6 (**Supplementary Fig. 4e, f**).

3. In 3a HGF treatment leads to increased PTK6 in the cytoplasm; is it active, PY342 positive? Membrane fractions are not shown. Is HGF activating PTK6 signaling at the plasma membrane to promote oncogenesis?

Responses:

- a. We performed the **Fig.3a** experiments to detect PTPN22, β -catenin, active PTK6 (Y342) and PTK6 in immunoblots of nuclear, cytosol and membrane fractions.
- b. As shown in **Fig. 3a**, active p-PTK6 were significantly increased in cytosol and membrane fractions. We performed gene expressions (qRT-PCR), cell migration and invasion assay to provide evidences in promoting oncogenesis after HGF activating PTK6 signaling in **Supplemental Figure 4a-c**. Meanwhile, we further we performed RNA sequencing (RNA-Seq) of transcriptome followed by comparing gene signatures profiles of gene sets enrichment analysis (GSEA) in parental and HGF stimulation of Huh-7 cells. GSEA analysis confirmed upregulation of HGF-related pathway, as well as the activation of c-MET downstream pathway and also identified increased expression of Metastasis and EMT-related signatures in PTPN22/PTK6/ β -catenin downstream pathway (**Supplementary**

Figure 4d and Supplementary Table 2).

4. Several reports indicate oncogenic roles for active PTK6 at the plasma membrane. Zheng et al. 2012 showed that P-PTK6 at the plasma membrane induces cell shape changes shown in Fig. 2B. Here, immunofluorescence images of active PTK6 are not shown in any panel and membrane fractions are not included in the immunoblotting studies.

Responses:

As suggested, we performed immunofluorescence images of active PTK6 and total form of PTK6 in **Fig. 2f** and immunoblotting data of total and active PTK6 for nuclear, cytosol and membrane fractions in **Fig. 3a and 3c** in Huh-7 and SK-hep1 cells.

5. Fig. 1a: You cannot definitively identify proteins on a Coomassie stained gel without immunoblotting.

Responses:

We moved **original Fig. 1a to Supplementary Fig 1a** and performed immunoblotting of PSPC1, PSF, p54nrb and PTK6 in **new Fig. 1a.**

6. Fig. 1g. How does EMT marker expression look with overexpression of WT PSPC1 alone?

Responses:

According to our previous paper “PSPC1 is the contextual determinant of Smad2/3 targets to activate EMT stemness, proliferation and metastatic switch” published in Nature cell biology⁸, the wild-type PSPC1 overexpression could increase the expression N-cadherin, core EMT-TFs, and core stemness-TFs at both the RNA and protein levels in SK-Hep1 cells.

7. Fig 2a. This panel does not demonstrate that PTK6 phosphorylates Y523 as indicated in the text. It shows that PSPC1 Y523 is required for PSPC1-PTK6 association. The impact of the Y523F mutation on PTK6 mediated PSPC1 phosphorylation should be shown. Supplemental 6c shows that PSPC1 PY523 associates with PTK6, not that PTK6 phosphorylates PSPC1 at this residue.

Responses:

We performed *in vitro* kinase assays to validate that active recombinant

human PTK6 phosphorylates PSPC1, but not PSPC1-Y523F mutant, by detecting with 4G10 phospho-tyrosine specific antibody (Supplementary Fig. 2d). Moreover, we also performed PTK6 ADP-Glo™ Kinase Assay to measure the kinase ability of PTK6 in PSPC1 Y523F mutation and found that PTK6 could phosphorylate PSPC1 at Y523 residue in Supplemental Figure 6b.

8. Fig. 2b. Are these really spheroids and not clusters? The fluorescent cells do not appear to be part of the spheroids as indicated in the legend. It would be interesting to include staining for PSPC1, PTK6, and P-PTK6 in Fig. 2b.

Responses:

Thank you for the suggestions, we re-performed spheroids culture in phase image in the indicated PSPC1/PTK6 transfectants in SK-hep1 cells and displayed new figures in Fig. 2i. As suggested, we performed immunofluorescence staining for PSPC1, PTK6 and p-PTK6 in Fig. 2f.

9. In Fig. 6f, CT131 increases activation of nuclear PTK6 but not cytoplasmic PTK6 and there does not appear to be a shift in total PTK6. Immunofluorescence staining for active P-PTK6 and total PTK6 should be shown.

Responses:

As suggested, we performed immunofluorescence staining for active P-PTK6 and total PTK6 localization under CT131 treatment in Mahlavu cells in Supplemental Figure 7e.

10. There are several technical issues. Problems with the language are present throughout: a few examples, “A prometastatic activator upregulated-PSPC1 in cancers might emerge as the lethal metastasis,” “Indigenous knockout,” “we performed divergent PSPC1/PTK6 constructs,” etc. Typos are also present throughout such as r-catenin (γ -catenin). Inclusion of size markers on the westerns is recommended. Quantitation of protein expression changes highlighted in the westerns would be helpful. Figures: 1a, d could be shown in supplemental data. TOP/FOP Flash constructs in Fig. 3e and f are not required. Remove 3e and f construct figure.

Responses:

a. We send our manuscript for English editing via Springer Nature

Author Services as shown by the enclosed Certificate.

- b. We added size markers onto all the immunoblotting data.
- c. We move **Figs. 1a and 1d** to **Supplemental Figures 1a and 1b**.
- d. We removed TOP/FOP flash constructs in **Figs. 3f and 3g**.

Reviewer #2 (Remarks to the Author):

Lang et al. reported interesting aspects of PSPC1 functions. The authors first identified PTK6, which has tumor suppressive functions, as one of the major PSPC1-interacting proteins. They examined the importance of this interaction in cancer by using cell-based and mouse cancer models as well as cancer clinical data. Then they have shown regulatory roles of PSPC1 in the PTK6/beta-catenin axis. However, the drawback of this manuscript is to contain numerous grammatical errors and typos. In addition, one of the major conclusions is not fully supported by the data shown in the manuscript as described below.

Major points

1. The authors should consider consulting a language editing service. There are numerous grammatical errors, typos, and uncommon usage of terms in the manuscript. Thus, there are many ambiguous sentences, which often disturb reading. Even in the abstract, multiple sentences are not clear, and thus it would be possible not to convey the authors' thoughts to readers.

Responses:

Thanks, we send our manuscript for English editing via Springer Nature Author Services as shown by the enclosed Certificate.

2. The authors claim that PTK6-CT131 is a dual inhibitor targeting oncogenic PTK6 and PTK6, and the interaction between the CT-131 and endogenous PTK6 is shown in the model of Figure 8. But, no biochemical data for this interaction is shown in the manuscript. Previous studies have shown that PTK6 dimerize and oligomerize with DBHS family proteins via the NOPS domain and the coiled-coil domain, respectively, but C-terminal domain is not reported for contributing the interaction. The PTK6-CT131 alone actually binds PTK6 (and Y523F mutant)?

Responses:

For answering these questions, we added multiple new figures and modified our description in **Pages 17-18**.

- a. Our results so far demonstrated that the C-terminal domain of PSPC1 (PSPC1-CT131) served as a molecular docking target to PSPC1 and PTK6 (Fig. 6a) to modulate oncogenic subcellular translocations and facilitate tumor progression in HCC. With unique proline-rich interacting domain of PSPC1-CT131, we hypothesized that PSPC1-CT131 might simultaneously target PSPC1 and SH3 domain29 of PTK6 (**Supplementary Fig. 2c and Supplementary Fig. 7a**). We also performed that ectopic expression of CT131 could co-localize with endogenous PSPC1 and associated with PSPC1 in Mahlavue cells (**Supplementary Figures 7d and 7f**).
- b. We further performed co-IP assay to examine the interaction of PSPC1-CT131 could both bind to wild type and Y523F mutant of PSPC1 in SK-hep1 in **Supplemental Figures 7g**.

3. Related to the above point, how does PSPC1-CT131 inhibit endogenous PSPC1 functions? In Figure 6h, PSPC1 expression was reduced by expression of EGFP-PSPC1-CT131. What is an interpretation for this result? Is this possible to be related to the mechanism of PSPC1 inhibition by the CT-131?

Responses:

In **Fig. 6h**, we did observe PSPC1-CT131-treatment could interact and reduce expression of endogenous PSPC1 in a dose dependent manner. This observation is validated by immunofluorescence experiment in **Supplementary Fig. 7d**. The inhibitory effects of PSPC1-CT131 might hijack the endogenous PSPC1 to increase PSPC1 protein degradation and/or reduce PSPC1 transcription. More detailed experiments will be required for future optimization of PSPC1-CT131 as a therapeutic agent.

Minor points

1. Fig. 1c: The labels of immunoblotting should be corrected to HA or FLAG as used in Figure S1b and S1c. (Input section)

Responses:

We corrected those figures. Thanks!

2. In Figure S2a, in vitro kinase assay using PTK6 and PSPC1 WT is performed. The PSPC1-Y523F mutant is actually not phosphorylated

in this assay?

Responses:

We further performed the PSPC1-WT and PSPC1-Y523F mutant examined by *in vitro* kinase assay and confirmed that PSPC1-Y523F were actually not phosphorylated by PTK6 in **Supplementary Fig. 2d.**

3. In middle cartoon of Figure 8, cytoplasmic PTK6 should be phosphorylated according to the data shown in Figure 3c?

Responses:

Corrected. Thanks!

4. Methods: In “Antibodies” section, information of cell lines and plasmids are included. Please fix this point.

Responses:

Corrected. Thanks!

5. Methods: Transfection methods are described in “PSPC1 site-directed mutagenesis” section. The transfection methods should be separated or combined with other sections.

Responses:

Corrected. Thanks!

6. In Figure 5b & c and Figure 6 c & d, no labels are found on the x-axis. I guess that these are same as in Figure 5d and Figure 6e, respectively. But, I think that this labeling is uncommon. Please fix this point.

Responses:

Corrected. Thanks!

Reviewer #3 (Remarks to the Author):

General comment: PTK6 kinase plays an opposite role in tumor progression which is dependent on its subcellular localization and transportation between nucleus and cytoplasm. PSPC1 protein, a substrate of PTK6 kinase closed to splicing speckles and recently has been reported by Dr. Jou’s group to regulate TGF- β signal pathway for pro-metastatic switches of cancer cells. In this study, Lang et al. showed that PSPC1-Y523F cause oncogenic subcellular translocations of

cytoplasmic PTK6 and nuclear β -catenin as well as Wnt3a autocrine loop for promoting liver cancer progression. PSPC1-CT131 which is the proline-rich C-terminal domain of PSPC1 containing NLS (nuclear localization signal), interacts with PTK6 in nucleus to inhibit PSPC1/PTK6 oncogenic functions. The study is interesting and provides mechanistic insights for understanding the role of subcellular localization of PSPC1/PTK6/ β -catenin in regulating liver cancer progression. However, some key experiments supporting individual claims may not be sufficient to reveal causal and concrete regulatory mechanisms. Several issues should be clarified.

Comment 1: The SH2 domain interacts with phosphorylated tyrosine residues and SH3 domain binds to proline-rich region (Yeatman T. J., 2004). Does proline-rich region of PSPC1 directly bind to SH3 domain of PTK6 (Supplementary Figure 1b, c)?

Responses:

We performed additional co-IP assays for PTK6-SH3 domain with PTK6 deletion constructs and PSPC1 proline-rich region deletion construct. (Supplementary Figures 1f, g). We validated that proline-rich region of PSPC1 directly bind to SH3 domain of PTK6.

Comment 2: The author claims that PSPC1 upregulation and PSPC1-Y523F released PTK6 nuclear sequestration; however, the existing evidence of this study for demonstrating the above results are not enough. In particular, the interaction between PSPC1 and PTK6 should be confirmed in different cellular fractions and the immunofluorescent images which indicate PSPC1-Y523F instead of PSPC1-wild type released PTK6 nuclear sequestration should be provided.

Responses:

Thank you for the suggestions. We performed PSPC1 and PTK6 interaction in different cellular fraction (nuclear and cytosol) in SK-hep1 cells (Figs. 3d and 3e). We also performed immunofluorescence experiments to indicate PSPC1-Y523F instead of PSPC-wild-type released PTK6 nuclear sequestration as shown in Fig. 2f.

Comment 3: For demonstrating the interaction between PSPC1 and PTK6 and their subcellular translocation, almost experiments in this study were performed by ectopic gene expression system. To obtain more

convincing evidence that PSPC1 interacts with PTK6 naturally, endogenous gene expression experiments should be performed.

Responses:

We performed endogenous co-IP assays and showed that PSPC1 could associate with PTK6 endogenously in Huh-7 cells in Fig. 1a and 1b.

Comment 4: Which kinase(s) phosphorylates PTK6?

Responses:

The hepatocyte growth factor (HGF) and its specific receptor c-Met tyrosine kinase regulate cancer cell migration, thereby conferring an aggressive phenotype^{2,6}. Our results demonstrated that HGF-treated Huh 7 cells not only activated profiles of EMT and stemness but also c-Met signaling pathway by using transcriptome and GSEA analysis. IP-Western analysis of c-Met and PTK6 interaction with and without treatments of c-Met inhibitor XL-184 implicated that c-Met is the kinase to phosphorylate pY342-PTK6 to increase its oncogenic functions (Supplementary Figures 4d-f).

Comment 5: In Supplementary Figure 5a IHC analysis, the expression of PTK6 in PSPC1-Y523F/PTK6 is lower than PSPC1/PTK6 group. How does mutant PSPC1 regulate PTK6 expression?

Responses:

It is our mistake. We have replaced the correct IHC data in Supplementary Figure 5a.

Comment 6: Supplementary Figure 7c, for clearly confirming the localization and expression of EGFP, EGFP images should be shown as an independent panel. Moreover, both N-cadherin and E-cadherin are cell-cell adhesion proteins, their localization should be at the cell-cell contacts.

Responses:

As suggested, we added the independent panel images of EGFP in Fig. 6b. Meanwhile, we corrected the immunofluorescent images of N-cadherin and E-cadherin in Supplementary Figure 7c.

Comment 7: The evidence for supporting PTK6/PSPC1 to regulate Wnt3a autocrine may not be sufficient. Detection of Wnt3a should in animal experiments and clinical samples is recommended.

Responses:

As suggested, we detected Wnt3a expression by using IHC and examined the expression percentage of Wnt3a of human HCC tumor samples (Figs. 5a and 5e) and tumors isolated from xenograft and metastatic tumors (Supplementary Figures 5a and 5b).

Minor points:

1. For Figure 1h, 1k, 2d, 2f and 6d, the labeling of y-axis should be corrected to “Invasive cells per field”.

Responses:

Thanks! We have modified accordingly.

2. In Supplementary Figure 1b, the author established a GFP-PSPC1-CT construct to confirm that Proline-rich C-terminal domain is a major binding site on PSPC1 for PTK6 binding. However, the full-length of PSPC1 is cloned in a HA vector. Both PSPC1-CT and full-length of PSPC1 should be individually cloned in the same vector.

Responses:

Actually, we established HA-tagged and GFP-tagged constructs for both full-length and PSPC1-CT131 of PSPC1. We obtained similar results from either HA-or GFP-tag. When examining the effects of PSPC1-CT131 on the PSPC1 interaction and expressions, we will use different tag labeling to avoid cross-reaction.

3. For Figure 3b immunoprecipitation assay, the input of β -catenin, PSPC1 and PTK6 as well as internal control such as β -actin should be shown.

Responses:

We added the input (TCL) of β -catenin, PSPC1 and PTK6 as well as the internal control of β -actin in Fig. 3b.

4. For Supplementary 4a, the groups of cell morphology in 2D culture are not labeled.

Responses:

Thanks! We have corrected.

5. The spheroid images of Figure 2b are not clear.

Responses:

Thanks! We replaced the spheroid images in **Fig. 2i**.

References in the Responses:

- 1 Goel, R. K. & Lukong, K. E. Tracing the footprints of the breast cancer oncogene BRK - Past till present. *Biochim Biophys Acta* **1856**, 39-54, doi:10.1016/j.bbcan.2015.05.001 (2015).
- 2 Locatelli, A., Lofgren, K. A., Daniel, A. R., Castro, N. E. & Lange, C. A. Mechanisms of HGF/Met signaling to Brk and Sam68 in breast cancer progression. *Horm Cancer* **3**, 14-25, doi:10.1007/s12672-011-0097-z (2012).
- 3 Palka-Hamblin, H. L. *et al.* Identification of β -catenin as a target of the intracellular tyrosine kinase PTK6. *Journal of Cell Science* **123**, 236-245, doi:10.1242/jcs.053264 (2010).
- 4 Kikuchi, I. *et al.* Dephosphorylated parafibromin is a transcriptional coactivator of the Wnt/Hedgehog/Notch pathways. *Nat Commun* **7**, 12887, doi:10.1038/ncomms12887 (2016).
- 5 Tang, C., Takahashi-Kanemitsu, A., Kikuchi, I., Ben, C. & Hatakeyama, M. Transcriptional Co-activator Functions of YAP and TAZ Are Inversely Regulated by Tyrosine Phosphorylation Status of Parafibromin. *iScience* **1**, 1-15, doi:10.1016/j.isci.2018.01.003 (2018).
- 6 Castro, N. E. & Lange, C. A. Breast tumor kinase and extracellular signal-regulated kinase 5 mediate Met receptor signaling to cell migration in breast cancer cells. *Breast Cancer Res* **12**, R60, doi:10.1186/bcr2622 (2010).
- 7 Regan Anderson, T. M. *et al.* Breast tumor kinase (Brk/PTK6) is a mediator of hypoxia-associated breast cancer progression. *Cancer Res* **73**, 5810-5820, doi:10.1158/0008-5472.CAN-13-0523 (2013).
- 8 Yeh, H.-W. *et al.* PSPC1 mediates TGF- β 1 autocrine signalling and Smad2/3 target switching to promote EMT, stemness and metastasis. *Nature Cell Biology* **20**, 479-491, doi:10.1038/s41556-018-0062-y (2018).

Reviewers' comments:

Reviewer #1 (Remarks to the Author):

The authors have addressed many of the concerns raised in the previous review. However, several questions remain. The splicing regulator PSF binds both PTK6 (Lukong, Richard, 2009) and PSPC1 (co-immunoprecipitation in Fig. 1a), and PSF associates with PTK6 through SH3 domain-polyproline interaction and is also a PTK6 substrate. Potential repercussions of disrupting PTK6-PSF association by overexpressing PSPC1 or PSPC1-CT131 are not discussed. The impact of PSPC1 overexpression and knockdown on endogenous PTK6 localization and activity is not demonstrated. In addition, PTK6 is not overexpressed alone, but only in combination with PSPC1 constructs, so it is difficult to differentiate distinct functions for PTK6 and PSPC1.

Figure 1d: How are PTK6 WT and KM localized within the cell without the overexpression of PSPC1? It would be interesting to also include the membrane fraction.

Figure 2f: PSPC1 and PTK6 + PSPC1 are expressed by transfection. PTK6 should also be transfected alone, and both total and phospho-PTK6 localization should be shown without cotransfection of PSPC1 constructs.

The SK-Hep-1 cell line used in several experiments (Fig. 2f, 2g, 3c etc.) appears to express endogenous PTK6, at levels not so different from the SNU387 "high" PTK6 expresser in Fig. S1 h, but we do not see any endogenous PTK6 by IF or western. Several liver cancer cell lines coexpress both PSPC1 and PTK6. The data would be stronger if the impact of PSPC1 on endogenous PTK6 localization/activation was examined.

The Fig. 2g legend states that PTK6 suppressed PSPC1-potentiated EMT. HA-PSPC1 is not expressed without PTK6 in this figure, so one cannot tell. With coexpressed wild type PSPC1, total and active PTK6 are nuclear (Fig. 2f), and nuclear PTK6 has been shown to inhibit beta-catenin transcription (Palka-Hamblin et al. 2010) and cell growth (Brauer, Zheng et al., 2010). PTK6 localization is different in the Y523F PSPC1 expressing cells; active PTK6 is at the plasma membrane (Fig. 2f). Membrane-associated active PTK6 has been shown to activate beta-catenin regulated transcription (Palka-Hamblin et al. 2010), and to induce the EMT (Zheng, Can Res, 2013).

SK-hep1 transfectant orthotopic tumor studies with cells expressing PTK6 alone have not been included, so it is impossible to ascertain the role of PTK6 without overexpressed PSPC1.

Immunofluorescence for total and p-PTK6 should be shown in Fig. 6b, not in supplemental data.

In Fig. 6f similar levels of total PTK6 are present in the cytoplasmic and nuclear fractions in cells expressing either the PSPC1-CT131 or MutNLS-CT, not supporting the authors' assertion that expression of PSPC1-CT, but not MutNLS-CT altered subcellular localization of PTK6. While total protein levels do not appear to change, PSPC1-CT131 expression does result in increased active p-PTK6 in the nucleus (Fig. 6c nucleus). Do PSPC1 and PSPC1-CT131 sequester PTK6 in the nucleus or transport a pool of p-PTK6 to the nucleus? Or activate PTK6 in the nucleus through binding its SH3 domain relieving intramolecular inhibition?

Overexpression of CT131 would compete with other interactions of the PTK6 SH3 domain, such as its ability to interact with PSF. Potential contributions of PSF which binds both PSPC1 and PTK6 and is coimmunoprecipitated with PSPC1 in Fig. 1a are not discussed.

More detail should be provided in the figure legend of the model presented in Fig. 8. It should be noted that many functions attributed to PSC1 in the figure have been previously demonstrated for PTK6, including regulation of beta-catenin, cell migration and the EMT.

There are still some grammatical/spelling mistakes throughout the manuscript. In Fig. 2f legend, the meaning of "in divergent" is unclear? Fig. S1 h. Mahlavu or Mahalvu?

Reviewer #2 (Remarks to the Author):

After the author's revision, I still notice multiple issues such as lack of controls in the experiments. In addition, I have to claim that the mechanisms of actions of PSPC1-CT131 remains still ambiguous even though it is an integral part of this manuscript. Our comments on the authors' responses are shown below (start from arrows).

Reviewer #2 (Remarks to the Author):

Lang et al. reported interesting aspects of PSPC1 functions. The authors first identified PTK6, which has tumor suppressive functions, as one of the major PSPC1-interacting proteins. They examined the importance of this interaction in cancer by using cell-based and mouse cancer models as well as cancer clinical data. Then they have shown regulatory roles of PSPC1 in the PTK6/beta-catenin axis. However, the drawback of this manuscript is to contain numerous grammatical errors and typos. In addition, one of the major conclusions is not fully supported by the data shown in the manuscript as described below.

Major points

1. The authors should consider consulting a language editing service. There are numerous grammatical errors, typos, and uncommon usage of terms in the manuscript. Thus, there are many ambiguous sentences, which often disturb reading. Even in the abstract, multiple sentences are not clear, and thus it would be possible not to convey the authors' thoughts to readers.

Responses:

Thanks, we send our manuscript for English editing via Springer Nature Author Services as shown by the enclosed Certificate.

-> OK, but if you added anything after the English editing, they should be checked again by the editing service.

2. The authors claim that PSPC1-CT131 is a dual inhibitor targeting oncogenic PSPC1 and PTK6, and the interaction between the CT-131 and endogenous PSPC1 is shown in the model of Figure 8. But, no biochemical data for this interaction is shown in the manuscript. Previous studies have shown that PSPC1 dimerize and oligomerize with DBHS family proteins via the NOPS domain and the coiled-coil domain, respectively, but C-terminal domain is not reported for contributing the interaction. The PSPC1-CT131 alone actually binds PSPC1 (and Y523F mutant)?

Responses:

For answering these questions, we added multiple new figures and modified our description in Pages 17-18.

a. Our results so far demonstrated that the C-terminal domain of PSPC1 (PSPC1-CT131) served as a molecular docking target to PSPC1 and PTK6 (Fig. 6a) to modulate oncogenic subcellular translocations and facilitate tumor progression in HCC. With unique proline-rich interacting domain of PSPC1-CT131, we hypothesized that PSPC1-CT131 might simultaneously target PSPC1 and SH3 domain²⁹ of PTK6 (Supplementary Fig. 2c and Supplementary Fig. 7a). We also performed that ectopic expression of CT131 could co-localize with endogenous PSPC1 and associated with PSPC1 in Mahlavue cells (Supplementary Figures 7d and 7f).

b. We further performed co-IP assay to examine the interaction of PSPC1-CT131 could both bind to wild type and Y523F mutant of PSPC1 in SK-hep1 in Supplemental Figures 7g.

-> In Supplementary Figure 7d, since no merged images are shown, it is hard to judge if they are co-localized. It would be better to show co-localization by using line profiles. In Supplementary

Figure 7f, the input data of the IP should be shown. In Supplementary Figure 7g, negative controls should be included. The authors claim that GFP-PSPC1-CT binds both PSPC1 WT and Y523F, however, it cannot rule out the possibility that the beads and/or antibodies used bind PSPC1 WT and Y523F. Thus, negative controls are essential.

3. Related to the above point, how does endogenous PSPC1 functions? In Figure 6h, PSPC1 expression was reduced by expression of EGFP-PSPC1-CT131. What is an interpretation for this result? Is this possible to be related to the mechanism of PSPC1 inhibition by the CT-131?

Responses:

In Fig. 6h, we did observe PSPC1-CT131-treatment could interact and reduce expression of endogenous PSPC1 in a dose dependent manner. This observation is validated by immunofluorescence experiment in Supplementary Fig. 7d. The inhibitory effects of PSPC1-CT131 might hijack the endogenous PSPC1 to increase PSPC1 protein degradation and/or reduce PSPC1 transcription. More detailed experiments will be required for future optimization of PSPC1-CT131 as a therapeutic agent.

-> The authors observed reduction of PSPC1 by the expression of PSPC1-CT, although the mechanisms are unclear. The authors claim that this observation is validated by the immunofluorescence data. However, I am still wondering how the PSPC1 antibody did not cross-react with PSPC1-CT131? Which part is the epitope of the antibody in PSPC1? It would be also important to show quantification data of fluorescent intensities of PSPC1 using quantified multiple images and performing statistical analyses.

The authors mentioned several possible mechanisms for the actions of PSPC1-CT131. I think the authors should provide some experimental data to support their hypothesis. At present, it is even extremely unclear if the mechanism is direct or indirect. Thus, the model shown in Figure 8c would contain too much analogy.

Minor points

1. Fig. 1c: The labels of immunoblotting should be corrected to HA or FLAG as used in Figure S1b and S1c. (Input section)

Responses:

We corrected those figures. Thanks!

-> The label looks still strange. In the previous version, the IPs were performed with STK6. However, in the revised version it is replaced by HA. If HA antibody was used for the IP, it does not fit the immunoblot data. The authors should precisely describe the data and carefully prepare the figures.

2. In Figure S2a, in vitro kinase assay using PTK6 and PSPC1 WT is performed. The PSPC1-Y523F mutant is actually not phosphorylated in this assay?

Responses:

We further performed the PSPC1-WT and PSPC1-Y523F mutant examined by in vitro kinase assay and confirmed that PSPC1-Y523F were actually not phosphorylated by PTK6 in Supplementary Fig. 2d.

-> There is no negative control data for this kinase assay. PTK(-) controls should be required. In addition, phosphatase treatment would be beneficial to show that the signals are phosphorylated proteins.

3. In middle cartoon of Figure 8, cytoplasmic PTK6 should be phosphorylated according to the data shown in Figure 3c?

Responses:
Corrected. Thanks!

-> OK.

4. Methods: In "Antibodies" section, information of cell lines and plasmids are included. Please fix this point.

Responses:
Corrected. Thanks!

-> The title of the "Proteins tested by antibodies and characteristics of the corresponding antibodies" in supplementary table should be edited. And in the table, the boundaries of the target antibodies are unclear, especially between P53 and PTEN. Lines should be put.

5. Methods: Transfection methods are described in "P53 site- directed mutagenesis" section. The transfection methods should be separated or combined with other sections.

Responses:
Corrected. Thanks!

-> Transfection methods are described in the method section "Cell culture and transfection". The title should be changed (e.g., to "Cell culture, plasmids, and transfection". The reference numbers in the method section should be shown with superscript font style.

6. In Figure 5b & c and Figure 6c & d, no labels are found on the x-axis. I guess that these are same as in Figure 5d and Figure 6e, respectively. But, I think that this labeling is uncommon. Please fix this point.

Responses:
Corrected. Thanks!

-> OK.

Reviewer #3 (Remarks to the Author):

The authors adequately responded to all the questions I raised in the previous version of manuscript and the current version is much improved and comprehensive. I have no further comment.

Reviewer #1 (Remarks to the Author):

The authors have addressed many of the concerns raised in the previous review. However, several questions remain. The splicing regulator PSF binds both PTK6 (Lukong, Richard, 2009) and PSPC1 (co-immunoprecipitation in Fig. 1a), and PSF associates with PTK6 through SH3 domain-polyproline interaction and is also a PTK6 substrate. Potential repercussions of disrupting PTK6-PSF association by overexpressing PSPC1 or PSPC1-CT131 are not discussed. The impact of PSPC1 overexpression and knockdown on endogenous PTK6 localization and activity is not demonstrated. In addition, PTK6 is not overexpressed alone, but only in combination with PSPC1 constructs, so it is difficult to differentiate distinct functions for PTK6 and PSPC1.

Figure 1d: How are PTK6 WT and KM localized within the cell without the overexpression of PSPC1? It would be interesting to also include the membrane fraction.

Response:

As shown in Rebuttal Figure R1, we performed the ectopic expression of PTK6 or PTK6-KM (kinase dead) mutant in PSPC1-deficient SK-hep1 cells and detected subcellular localizations of PTK6 and p-PTK6 by Western blotting analysis. Although the total form PTK6 is evenly distributed in different subcellular localizations, the active p-PTK6 is accumulated more in cytosol and membrane fractions. Consistent to dynamic modulations of PSPC1/PTK6 in our study, expression of active p-PTK6 in cytosol and membrane fractions might be due to lack of PSPC1/PTK6 nuclear sequestration that led to cytoplasmic and oncogenic PTK6 and tumor growth in mice HCC orthotopic model. (Consistent to the results in Rebuttal Figures R2 and R3).

Figure 2f: PSPC1 and PTK6 + PSPC1 are expressed by transfection. PTK6 should also be transfected alone, and both total and phospho-PTK6 localization should be shown without cotransfection of PSPC1 constructs.

Response:

As shown in Rebuttal Figure R2, we expressed PTK6 in PSPC1-deficient SK-hep1 cells and detected subcellular localizations of total PTK6 and p-PTK6 by immunofluorescence (IF) analysis. Consistent to the results in Rebuttal Figure R1, total PTK6 might distribute evenly in different subcellular fractions but p-PTK6 is expressed more focusing on locations of cytoplasm and cell membrane.

The SK-Hep-1 cell line used in several experiments (Fig. 2f, 2g, 3c etc.) appears to express endogenous PTK6, at levels not so different from the SNU387 “high” PTK6

expresser in Fig. S1 h, but we do not see any endogenous PTK6 by IF or western. Several liver cancer cell lines coexpress both PSPC1 and PTK6. The data would be stronger if the impact of PSPC1 on endogenous PTK6 localization/activation was examined.

Response:

To illustrate the impact of PSPC1 and PSPC1-Y523F on endogenous PTK6 localization/activation, we expressed PSPC1 constructs into SNU-387 cells to survey its impact on endogenous PTK6 localizations by western blotting analysis. Consistent with results in main Figures, expression of upregulated-PSPC1, especially PSPC1-Y523F, led to increase expression of p-PTK6 in cytosol and membrane fractions and to reduce p-PTK6 in the nucleus (new subfigure of Supplementary Figure 4g).

The Fig. 2g legend states that PTK6 suppressed PSPC1-potentiated EMT. HA-PSPC1 is not expressed without PTK6 in this figure, so one cannot tell. With coexpressed wild type PSPC1, total and active PTK6 are nuclear (Fig. 2f), and nuclear PTK6 has been shown to inhibit beta-catenin transcription (Palka-Hamblin et al. 2010) and cell growth (Brauer, Zheng et al., 2010). PTK6 localization is different in the Y523F PSPC1 expressing cells; active PTK6 is at the plasma membrane (Fig. 2f). Membrane-associated active PTK6 has been shown to activate beta-catenin regulated transcription (Palka-Hamblin et al. 2010), and to induce the EMT (Zheng, Can Res, 2013).

Response:

In Fig. 2, we emphasize on the studies of PTK6 phosphorylation site PSPC1-Y523 in related to PTK6 nuclear sequestration and oncogenic functions. In Fig. 3, we further illustrate dynamic interactions of upregulated-PSPC1 or PSPC1-Y523F with PTK6 modulate Wnt/ β -catenin signaling. In both Figures plus Supplementary Fig. 3 and Fig. 4, we demonstrated dynamic modulations of subcellular translocations of PSPC1/PTK6/ β -catenin in related to EMT and oncogenic functions. Our results indeed suggested that nuclear PTK6 occupied PSPC1 to prevent PSPC1/ β -catenin-mediated EMT in HCC cells. In addition, active p-PTK6 translocated to cytoplasm and cell membrane when PSPC1-Y523F or upregulation of PSPC1 were expressed in HCC cells to activate PSPC1/Wnt/ β -catenin signaling pathway and HCC tumor progression.

SK-hep1 transfectant orthotopic tumor studies with cells expressing PTK6 alone have not been included, so it is impossible to ascertain the role of PTK6 without overexpressed PSPC1.

Response:

As shown in Rebuttal Fig. R3, expression of PTK6 in PSPC1-deficient SK-hep1 cells led to slightly increase of HCC tumor growth (n=6 for each group). The results are consistent that expression of PTK6 in PSPC1-deficient SK-hep1 cells could increase active p-PTK6 expression in cytosol and membrane (Rebuttal Fig. R2) to mediate tumor growth.

Immunofluorescence for total and p-PTK6 should be shown in Fig. 6b, not in supplemental data.

Response:

As suggested, we moved the results that “PSPC1-CT131, but not MutNLS-CT131, co-localized PSPC1 and p-PTK6 in nucleus via immunofluorescent analysis” to Fig 6b. Thanks!

In Fig. 6f similar levels of total PTK6 are present in the cytoplasmic and nuclear fractions in cells expressing either the PSPC1-CT131 or MutNLS-CT, not supporting the authors' assertion that expression of PSPC1-CT, but not MutNLS-CT altered subcellular localization of PTK6. While total protein levels do not appear to change, PSPC1-CT131 expression does result in increased active p-PTK6 in the nucleus (Fig. 6c nucleus). Do PSPC1 and PSPC1-CT131 sequester PTK6 in the nucleus or transport a pool of p-PTK6 to the nucleus? Or activate PTK6 in the nucleus through binding its SH3 domain relieving intramolecular inhibition?

Response:

Since PSPC1 and PSPC1-CT131 mainly expressed in nucleus, the possibility to act as a chaperon to transport p-PTK6 to the nucleus is unlikely. Therefore, we suggest that PSPC1 or PSPC1-CT131 could sequester active p-PTK6 in the nucleus through binding its SH3 domain.

Overexpression of CT131 would compete with other interactions of the PTK6 SH3 domain, such as its ability to interact with PSF. Potential contributions of PSF which binds both PSPC1 and PTK6 and is coimmunoprecipitated with PSPC1 in Fig. 1a are not discussed.

Response:

PSF, a PTK6-interacting protein and substrate, interacted with PTK6 through PTK6-SH3 domain. The c-terminal tyrosine residue of PSF phosphorylated by PTK6 promoted the PSF cytoplasmic re-localization. In contrast, PSPC1-CT131 interact with PTK6-SH3 domain in the nucleus but not interacted with PSF in the cytosol. We added a paragraph in Discussion section p.22.

More detail should be provided in the figure legend of the model in presented in Fig. 8. It should be noted that many functions attributed to PSCP1 in the figure have been previously demonstrated for PTK6, including regulation of beta-catenin, cell migration and the EMT.

Response:

We modified the legend of the proposed model in Figure 8 as suggested.

There are still some grammatical/spelling mistakes throughout the manuscript. In Fig. 2f legend, the meaning of “in divergent” is unclear? Fig. S1 h. Mahlavu or Mahalvu?

Response:

- a. We corrected the legend statement to “Immunofluorescence for the detection of subcellular localizations of PSCP1, p-PTK6 and PTK6 in SK-Hep1” in Figure 2f.**
- b. Mahlavu is corrected in Supplementary Figure 1h.**
- c. We send the manuscript to a senior professor for Scientific Editing. We also send the manuscript to the AJE for the English Editing (certificate enclosed).**

Reviewer #2 (Remarks to the Author):

After the author’s revision, I still notice multiple issues such as lack of controls in the experiments. In addition, I have to claim that the mechanisms of actions of PSCP1-CT131 remains still ambiguous even though it is an integral part of this manuscript. Our comments on the authors’ responses are shown below (start from arrows).

Reviewer #2 (Remarks to the Author):

Lang et al. reported interesting aspects of PSCP1 functions. The authors first identified PTK6, which has tumor suppressive functions, as one of the major PSCP1-interacting proteins. They examined the importance of this interaction in cancer by using cell-based and mouse cancer models as well as cancer clinical data. Then they have shown regulatory roles of PSCP1 in the PTK6/beta-catenin axis. However, the drawback of this manuscript is to contain numerous grammatical errors and typos. In addition, one of the major conclusions is not fully supported by the data shown in the manuscript as described below.

Major points

1. The authors should consider consulting a language editing service. There are numerous grammatical errors, typos, and uncommon usage of terms in the manuscript. Thus, there are many ambiguous sentences, which often disturb reading. Even in the abstract, multiple sentences are not clear, and thus it would be possible not to convey the authors’ thoughts to readers.

Responses:

Thanks, we send our manuscript for English editing via Springer Nature Author Services as shown by the enclosed Certificate.

-> OK, but if you added anything after the English editing, they should be checked again by the editing service.

Response:

We have send the manuscript to a senior PI in our organization for Scientific English editing. We then send the manuscript to the AJE for the final English Editing (Certificate enclosed). Thanks!

2. The authors claim that PSPC1-CT131 is a dual inhibitor targeting oncogenic PSPC1 and PTK6, and the interaction between the CT-131 and endogenous PSPC1 is shown in the model of Figure 8. But, no biochemical data for this interaction is shown in the manuscript. Previous studies have shown that PSPC1 dimerize and oligomerize with DBHS family proteins via the NOPS domain and the coiled-coil domain, respectively, but C-terminal domain is not reported for contributing the interaction. The PSPC1-CT131 alone actually binds PSPC1 (and Y523F mutant)?

Responses:

For answering these questions, we added multiple new figures and modified our description in Pages 17-18.

a. Our results so far demonstrated that the C-terminal domain of PSPC1 (PSPC1-CT131) served as a molecular docking target to PSPC1 and PTK6 (Fig. 6a) to modulate oncogenic subcellular translocations and facilitate tumor progression in HCC. With unique proline-rich interacting domain of PSPC1-CT131, we hypothesized that PSPC1-CT131 might simultaneously target PSPC1 and SH3 domain²⁹ of PTK6 (Supplementary Fig. 2c and Supplementary Fig. 7a). We also performed that ectopic expression of CT131 could co-localize with endogenous PSPC1 and associated with PSPC1 in Mahlavue cells (Supplementary Figures 7d and 7f).

b. We further performed co-IP assay to examine the interaction of PSPC1-CT131 could both bind to wild type and Y523F mutant of PSPC1 in SK-hep1 in Supplemental Figures 7g.

-> In Supplementary Figure 7d, since no merged images are shown, it is hard to judge if they are co-localized. It would be better to show co-localization by using line profiles. In Supplementary Figure 7f, the input data of the IP should be shown. In Supplementary Figure 7g, negative controls should be included. The authors claim that GFP-PSPC1-CT binds both PSPC1 WT and Y523F, however, it cannot rule out the possibility that the beads and/or antibodies used bind PSPC1 WT and Y523F. Thus, negative controls

are essential.

Response:

- a. In combined with suggestions from reviewer #1 and #2, we moved the original Supplementary Fig 7d to Fig. 6b with addition of co-localization line profiles (middle panel) and the merged staining intensities (bottom panel) of the IF results for supporting the binding of GFP-PSPC1-CT with PSPC1 WT and Y523F.**
- b. As suggested, we added the input data of IP assay in Supplementary Fig. 7f.**
- c. To demonstrate interacting specificity in IP assay of Supplementary Figure 7g, the first lane “-/-“ served as negative control and the indication of IgG heavy chain versus PSPC1 (HA and GFP) indicated the specificity of IP assays.**

3. Related to the above point, how does endogenous PSPC1 functions? In Figure 6h, PSPC1 expression was reduced by expression of EGFP-PSPC1-CT131. What is an interpretation for this result? Is this possible to be related to the mechanism of PSPC1 inhibition by the CT-131?

Responses:

In Fig. 6h, we did observe PSPC1-CT131-treatment could interact and reduce expression of endogenous PSPC1 in a dose dependent manner. This observation is validated by immunofluorescence experiment in Supplementary Fig. 7d. The inhibitory effects of PSPC1-CT131 might hijack the endogenous PSPC1 to increase PSPC1 protein degradation and/or reduce PSPC1 transcription. More detailed experiments will be required for future optimization of PSPC1-CT131 as a therapeutic agent.

-> The authors observed reduction of PSPC1 by the expression of PSPC1-CT, although the mechanisms are unclear. The authors claim that this observation is validated by the immunofluorescence data. However, I am still wondering how the PSPC1 antibody did not cross-react with PSPC1-CT131? Which part is the epitope of the antibody in PSPC1? It would be also important to show quantification data of fluorescent intensities of PSPC1 using quantified multiple images and performing statistical analyses.

Response:

- a. Two commercial available PSPC1 antibodies purchased from SANTA CRUZ BIOTECHNOLOGY (N-16: sc-84576) and Sigma-Aldrich (N-terminal, SAB-4200068) were against N-terminal of PSPC1 in our IF experiments. In contrast, our homemade antibody is against C-terminal CT131 of PSPC1. Therefore, there is no cross-react issue.**
- b. The quantification data of fluorescent intensities of PSPC1/PSPC1-CT131 is shown in Fig. 6b bottom panel.**

->The authors mentioned several possible mechanisms for the actions of PSPC1-CT131. I think the authors should provide some experimental data to support their hypothesis. At present, it is even extremely unclear if the mechanism is direct or indirect. Thus, the model shown in Figure 8c would contain too much analogy.

Response:

In the manuscript, we provide experimental data of biochemical, molecular biological and immunofluorescent labeling results to emphasize that treatment of PSPC1-CT131 to cancer cells could sequester and suppress the oncogenic expression of PSPC1 and the reciprocal p-PTK6/ β -catenin subcellular translocations to diminish tumor growth and prolong life of HCC mice.

To provide additional experimental data regarding to the inhibitory mechanism of PSPC1-CT131, we performed NGS transcriptome and applied Ingenuity Pathway Analysis (IPA) and Gene Set Enrichment Analysis (GSEA) to identify distinct PSPC1-CT131 therapeutic signaling pathways. IPA and GSEA analysis indicated that treatment of PSPC1-CT131 activated eIF2 signaling, ribosome biogenesis and proteostasis pathways. Indeed, activation of phospho-eIF2 α are known to induce lethal endoplasmic reticulum stress responses led to cancer cells death and tumor suppression. We validated the activation of phospho-eIF2 α but downregulation of p-PTK6 and PSPC1 by Western blotting analysis and provided a hypothetic PSPC1-CT131 therapeutic model. More detail underlining mechanisms and optimization of PSPC1-CT131 therapeutic strategies as an anticancer agent will be addressed in our next article.

Minor points

1. Fig. 1c: The labels of immunoblotting should be corrected to HA or FLAG as used in Figure S1b and S1c. (Input section)

Responses:

We corrected those figures. Thanks!

-> The label looks still strange. In the previous version, the IPs were performed with STK6. However, in the revised version it is replaced by HA. If HA antibody was used for the IP, it does not fit the immunoblot data. The authors should precisely describe the data and carefully prepare the figures.

Response:

We corrected the labelling of subfigures in Fig. 1c.

2. In Figure S2a, in vitro kinase assay using PTK6 and PSPC1 WT is performed. The

PSPC1-Y523F mutant is actually not phosphorylated in this assay?

Responses:

We further performed the PSPC1-WT and PSPC1-Y523F mutant examined by in vitro kinase assay and confirmed that PSPC1-Y523F were actually not phosphorylated by PTK6 in Supplementary Fig. 2d.

-> There is no negative control data for this kinase assay. PTK(-) controls should be required. In addition, phosphatase treatment would be beneficial to show that the signals are phosphorylated proteins.

Response:

Thanks for your suggestions. We further performed the PTK6 negative controls and the phosphatase treatment followed by in vitro kinase assay analysis as shown in Supplementary Fig. 2d.

4. Methods: In “Antibodies” section, information of cell lines and plasmids are included. Please fix this point.

Responses:

Corrected. Thanks!

-> The title of the “Proteins tested by antibodies and characteristics of the corresponding antibodies” in supplementary table should be edited. And in the table, the boundaries of the target antibodies are unclear, especially between PSPC1 and PTK6. Lines should be put.

Response:

We corrected and modified the Supplementary Table 4. Thanks!

5. Methods: Transfection methods are described in “PSPC1 site- directed mutagenesis” section. The transfection methods should be separated or combined with other sections.

Responses:

Corrected. Thanks!

-> Transfection methods are described in the method section “Cell culture and transfection”. The title should be changed (e.g., to “Cell culture, plasmids, and transfection”. The reference numbers in the method section should be shown with superscript font style.

Response:

We corrected the method section. Thanks!

Rebuttal Fig. R1:

Reply to Reviewer#1 suggestion:

Q: Figure 1d: How are PTK6 WT and KM localized within the cell without the overexpression of PSPC1? It would be interesting to also include the membrane fraction.

Subcellular distribution of wild-type (WT) and kinase-dead (KM) mutant of flag-tagged PTK6 in SK-hep1 cells by Western blotting analysis. Sp1, α -tubulin and Na⁺/K⁺-ATPase were used as internal controls for nuclear, cytoplasmic and membrane fractions, respectively.

Rebuttal Fig. R2:

Reply to Reviewer#1 suggestion:

Q: Figure 2f: PSPC1 and PTK6 + PSPC1 are expressed by transfection. PTK6 should also be transfected alone, and both total and phospho-PTK6 localization should be shown without co-transfection of PSPC1 constructs.

Immunofluorescence analysis for the expression of phospho-PTK6 (p-PTK6) and PTK6. SK-hep1 (deficient of PSPC1) cells were transfected with the indicated plasmids. Colors are p-PTK6 and PTK6 in red, and nuclei with DAPI in blue. The scale bar represents 25 μ m.

Rebuttal Fig. R3:

Reply to Reviewer#1 suggestion:

Q: SK-hep1 transfectant orthotopic tumor studies with cells expressing PTK6 alone have not been included, so it is impossible to ascertain the role of PTK6 without overexpressed PSPC1.

Tumorigenesis of ectopic expression of PTK6 in SK-hep1 (PSPC1 deficient) transfectant bearing luciferase expression injected into mice as HCC orthotopic model measured by representative bioluminescence images. Data represent mean \pm SEM (n=6/group). All data statistics based on: *p < 0.05 by one-way ANOVA with Bartlett's test.

Rebuttal Fig. R4:

Reply to Reviewer#2 suggestion:

Q: The authors mentioned several possible mechanisms for the actions of PSpC1-CT131. I think the authors should provide some experimental data to support their hypothesis.

Transcriptome analysis of PSpC1-CT131 treated Mahlavu cells. **a** Top3 Canonical Pathway analysis by IPA software. **b** Significant gene sets categories from next generation sequencing (NGS) analysis identified using gene set enrichment analysis (GSEA) including Protein synthesis, eIF2 signaling in Mahlavu cells after PSpC1-CT131 treatment. ES, enrichment score; NES: normalized enrichment score in PSpC1-CT131 high as compared to control population. **c** Validation of PSpC1-CT131 activated p-eIF2α proteostasis pathway to suppress PSpC1 and p-PTK6 in PSpC1-CT131-treated Mahlavu cells analyzed by Western blotting analysis. **d** A hypothetic pathway that PSpC1-CT131 treatment could activate p-eIF2α proteostasis pathway to suppress PSpC1 and p-PTK6 synergized HCC tumor progression.

REVIEWERS' COMMENTS:

Reviewer #1 (Remarks to the Author):

Overall the manuscript is improved. The data clearly demonstrate that PSpC1 and PTK6 interact and regulate each other. It is interesting that there is a correlation between PSpC PY523 , nuclear PTK6 and better patient outcome. A few issues with data remain.

It would be useful to include the rebuttal Fig R2 in the Supplemental figures.

Figure 2i is described as spheroids. It is difficult to conclude that any spheroids are shown in the figure. Images look like cell clusters and clumps.

Some of the immunoblotting data are difficult to interpret. In Fig. 2h, multiple bands are present for several of the proteins, including PTK6. In Fig. 3B, total PTK6 in the IP blot appears to be running much smaller, as it is far from the IgG band, in contrast to several figures where it is very close to the IgG band. Size markers are not provided for the Fig. 3B TCL blots.

Reviewer #2 (Remarks to the Author):

The authors addressed the major and minor points of my original revision. Therefore, I think that the manuscript is now suitable for publication.

REVIEWERS' COMMENTS:

Reviewer #1 (Remarks to the Author):

Overall the manuscript is improved. The data clearly demonstrate that PSPC1 and PTK6 interact and regulate each other. It is interesting that there is a correlation between PSPC PY523 , nuclear PTK6 and better patient outcome. A few issues with data remain.

It would be useful to include the rebuttal Fig R2 in the Supplemental figures.

Response:

As suggested, we moved the rebuttal figure R2 into the supplemental figures 3a and added sentences in the main text.

Figure 2i is described as spheroids. It is difficult to conclude that any spheroids are shown in the figure. Images look like cell clusters and clumps.

Response:

We performed and repeated spheroids formation assays for more than six times in SK-hep1 cells. In addition, our images of spheroids in SK-hep1 were similar with the morphology of spheroids published in other papers^{1, 2}. Therefore, the spheroid images in figure 2i is more likely spheroids of SK-Hep1 cells but not for cell clusters and clumps.

Some of the immunoblotting data are difficult to interpret. In Fig. 2h, multiple bands are present for several of the proteins, including PTK6. In Fig. 3B, total PTK6 in the IP blot appears to be running much smaller, as it is far from the IgG band, in contrast to several figures where it is very close to the IgG band. Size markers are not provided for the Fig. 3B TCL blots

Response:

As suggested, we replaced the Figures 2h and 3b with their original figures to correct the problem. Meanwhile, we added the size markers in figure 3b TCL blots.

Reviewer #2 (Remarks to the Author):

The authors addressed the major and minor points of my original revision. Therefore, I think that the manuscript is now suitable for publication.

Response:

Many Thanks.

References:

1. Yeh H-W, Lee S-S, Chang C-Y, Hu C-M, Jou Y-S. Pyrimidine metabolic rate limiting enzymes in poorly-differentiated hepatocellular carcinoma are signature genes of cancer stemness and associated with poor prognosis. *Oncotarget* **8**, 77734-77751 (2017).
2. Cheng J-S, *et al.* The MAP3K7-mTOR Axis Promotes the Proliferation and Malignancy of Hepatocellular Carcinoma Cells. *Frontiers in Oncology* **9**, (2019).